# Theoretical Analysis of Self-Training with Deep Networks on Unlabeled Data

**Colin Wei & Kendrick Shen & Yining Chen & Tengyu Ma**
Department of Computer Science
Stanford University
Stanford, CA 94305, USA
`{colinwei,kshen6,cynnjjs,tengyuma}@stanford.edu`

## Abstract

Self-training algorithms, which train a model to fit pseudolabels predicted by an-
other previously-learned model, have been very successful for learning with unla-
beled data using neural networks. However, the current theoretical understanding
of self-training only applies to linear models. This work provides a unified theo-
retical analysis of self-training with deep networks for semi-supervised learning,
unsupervised domain adaptation, and unsupervised learning. At the core of our
analysis is a simple but realistic "expansion" assumption, which states that a low-
probability subset of the data must expand to a neighborhood with large probabil-
ity relative to the subset. We also assume that neighborhoods of examples in dif-
ferent classes have minimal overlap. We prove that under these assumptions, the
minimizers of population objectives based on self-training and input-consistency
regularization will achieve high accuracy with respect to ground-truth labels. By
using off-the-shelf generalization bounds, we immediately convert this result to
sample complexity guarantees for neural nets that are polynomial in the margin
and Lipschitzness. Our results help explain the empirical successes of recently
proposed self-training algorithms which use input consistency regularization.

## 1 Introduction

Though supervised learning with neural networks has become standard and reliable, it still often
requires massive *labeled* datasets. As labels can be expensive or difficult to obtain, leveraging
unlabeled data in deep learning has become an active research area. Recent works in semi-supervised
learning (Chapelle et al., 2010; Kingma et al., 2014; Kipf & Welling, 2016; Laine & Aila, 2016;
Sohn et al., 2020; Xie et al., 2020) and unsupervised domain adaptation (Ben-David et al., 2010;
Ganin & Lempitsky, 2015; Ganin et al., 2016; Tzeng et al., 2017; Hoffman et al., 2018; Shu et al.,
2018; Zhang et al., 2019) leverage lots of unlabeled data as well as labeled data from the same
distribution or a related distribution. Recent progress in unsupervised learning or representation
learning (Hinton et al., 1999; Doersch et al., 2015; Gidaris et al., 2018; Misra & Maaten, 2020;
Chen et al., 2020a;b; Grill et al., 2020) learns high-quality representations without using any labels.

Self-training is a common algorithmic paradigm for leveraging unlabeled data with deep networks.
Self-training methods train a model to fit pseudolabels, that is, predictions on unlabeled data made
by a previously-learned model (Yarowsky, 1995; Grandvalet & Bengio, 2005; Lee, 2013). Recent
work also extends these methods to enforce stability of predictions under input transformations such
as adversarial perturbations (Miyato et al., 2018) and data augmentation (Xie et al., 2019). These
approaches, known as input consistency regularization, have been successful in semi-supervised
learning (Sohn et al., 2020; Xie et al., 2020), unsupervised domain adaptation (French et al., 2017;
Shu et al., 2018), and unsupervised learning (Hu et al., 2017; Grill et al., 2020).

Despite the empirical successes, theoretical progress in understanding how to use unlabeled data
has lagged. Whereas supervised learning is relatively well-understood, statistical tools for reasoning
about unlabeled data are not as readily available. Around 25 years ago, Vapnik (1995) proposed
the transductive SVM for unlabeled data, which can be viewed as an early version of self-training,
yet there is little work showing that this method improves sample complexity (Derbeko et al., 2004).

Working with unlabeled data requires proper assumptions on the input distribution (Ben-David et al., 2008). Recent papers (Carmon et al., 2019; Raghunathan et al., 2020; Chen et al., 2020c; Kumar et al., 2020; Oymak & Gulcu, 2020) analyze self-training in various settings, but mainly for linear models and often require that the data is Gaussian or near-Gaussian. Kumar et al. (2020) also analyze self-training in a setting where gradual domain shift occurs over multiple timesteps but assume a small Wasserstein distance bound on the shift between consecutive timesteps. Another line of work leverages unlabeled data using non-parametric methods, requiring unlabeled sample complexity that is *exponential* in dimension (Rigollet, 2007; Singh et al., 2009; Urner & Ben-David, 2013).

This paper provides a unified theoretical analysis of self-training *with deep networks* for semi-supervised learning, unsupervised domain adaptation, and unsupervised learning. Under a simple and realistic expansion assumption on the data distribution, we show that self-training with input consistency regularization using a deep network can achieve high accuracy on true labels, using unlabeled sample size that is polynomial in the margin and Lipschitzness of the model. Our analysis provides theoretical intuition for recent empirically successful self-training algorithms which rely on input consistency regularization (Berthelot et al., 2019; Sohn et al., 2020; Xie et al., 2020).

Our expansion assumption intuitively states that the data distribution has good continuity within each class. Concretely, letting $P_i$ be the distribution of data conditioned on class $i$, expansion states that for small subset $S$ of examples with class $i$,

$$P_i(\text{neighborhood of } S) \geq cP_i(S) \tag{1.1}$$

where and $c > 1$ is the expansion factor. The neighborhood will be defined to incorporate data augmentation, but for now can be thought of as a collection of points with a small $\ell_2$ distance to $S$. This notion is an extension of the Cheeger constant (or isoperimetric or expansion constant) (Cheeger, 1969) which has been studied extensively in graph theory (Chung & Graham, 1997), combinatorial optimization (Mohar & Poljak, 1993; Raghavendra & Steurer, 2010), sampling (Kannan et al., 1995; Lovász & Vempala, 2007; Zhang et al., 2017), and even in early versions of self-training (Balcan et al., 2005) for the co-training setting (Blum & Mitchell, 1998). Expansion says that the manifold of each class has sufficient connectivity, as every subset $S$ has a neighborhood larger than $S$. We give examples of distributions satisfying expansion in Section 3.1. We also require a separation condition stating that there are few neighboring pairs from different classes.

Our algorithms leverage expansion by using input consistency regularization (Miyato et al., 2018; Xie et al., 2019) to encourage predictions of a classifier $G$ to be consistent on neighboring examples:

$$R(G) = \mathbb{E}_x[\max_{\text{neighbor } x'} \mathbf{1}(G(x) \neq G(x'))] \tag{1.2}$$

For unsupervised domain adaptation and semi-supervised learning, we analyze an algorithm which fits $G$ to pseudolabels on unlabeled data while regularizing input consistency. Assuming expansion and separation, we prove that the fitted model will denoise the pseudolabels and achieve high accuracy on the true labels (Theorem 4.3). This explains the empirical phenomenon that self-training on pseudolabels often improves over the pseudolabeler, despite no access to true labels.

For unsupervised learning, we consider finding a classifier $G$ that minimizes the input consistency regularizer with the constraint that enough examples are assigned each label. In Theorem 3.6, we show that assuming expansion and separation, the learned classifier will have high accuracy in predicting true classes, up to a permutation of the labels (which can't be recovered without true labels).

The main intuition of the theorems is as follows: input consistency regularization ensures that the model is locally consistent, and the expansion property magnifies the local consistency to global consistency within the same class. In the unsupervised domain adaptation setting, as shown in Figure 1 (right), the incorrectly pseudolabeled examples (the red area) are gradually denoised by their correctly pseudolabeled neighbors (the green area), whose probability mass is non-trivial (at least $c - 1$ times the mass of the mistaken set by expansion). We note that expansion is only required on the *population* distribution, but self-training is performed on the empirical samples. Due to the extrapolation power of parametric methods, the local-to-global consistency effect of expansion occurs *implicitly* on the population. In contrast, nearest-neighbor methods would require expansion to occur *explicitly* on empirical samples, suffering the curse of dimensionality as a result. We provide more details below, and visualize this effect in Figure 1 (left).

To our best knowledge, this paper gives the first analysis with polynomial sample complexity guarantees for deep neural net models for unsupervised learning, semi-supervised learning, and unsuper-

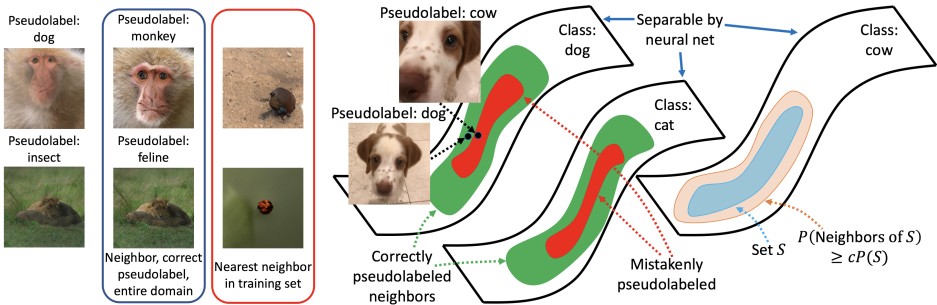

Figure 1: **Left: demonstrating expansion assumption.** Verifying the expansion assumption requires access to the population distribution and therefore we use the distribution generated by Big-GAN (Brock et al., 2018). We display typical examples of mistakenly classified images and their correctly classified neighbors, found by searching the *entire* GAN manifold (not just the training set). For contrast, we also display their nearest neighbors in the training set of 100K GAN images, which are much further away. This supports the intuition and assumption that expansion holds for the population set but not the empirical set. (More details are in Section E.1.) **Right: assumptions and setting for pseudolabeling.** For self-training with pseudolabels, the region of correctly pseudolabeled examples (in green) will be used to denoise examples with incorrect pseudolabels (in red), because by expansion, the green area will have a large mass which is at least $c - 1$ times the mass of the red area. As explained in the introduction, this ensures that a classifier which fits the pseudolabels and is consistent w.r.t. input transformations will achieve high accuracy on true labels.

vised domain adaptation. Prior works (Rigollet, 2007; Singh et al., 2009; Urner & Ben-David, 2013) analyzed nonparametric methods that essentially recover the data distribution exactly with unlabeled data, but require sample complexity exponential in dimension. Our approach optimizes parametric loss functions and regularizers, so guarantees involving the population loss can be converted to finite sample results using off-the-shelf generalization bounds (Theorem 3.7). When a neural net can separate ground-truth classes with large margin, the sample complexities from these bounds can be small, that is, polynomial in dimension.

Finally, we note that our regularizer $R(\cdot)$ corresponds to enforcing consistency w.r.t. adversarial examples, which was shown to be empirically helpful for semi-supervised learning (Miyato et al., 2018; Qiao et al., 2018) and unsupervised domain adaptation (Shu et al., 2018). Moreover, we can extend the notion of neighborhood in (1.1) to include data augmentations of examples, which will increase the neighborhood size and therefore improve the expansion. Thus, our theory can help explain empirical observations that consistency regularization based on aggressive data augmentation or adversarial training can improve performance with unlabeled data (Shu et al., 2018; Xie et al., 2019; Berthelot et al., 2019; Sohn et al., 2020; Xie et al., 2020; Chen et al., 2020a).

In summary, our contributions include: 1) we propose a simple and realistic expansion assumption which states that the data distribution has connectivity within the manifold of a ground-truth class 2) using this expansion assumption, we provide ground-truth accuracy guarantees for self-training algorithms which regularize input consistency on unlabeled data, and 3) our analysis is easily applicable to deep networks with polynomial unlabeled samples via off-the-shelf generalization bounds.

## 1.1 ADDITIONAL RELATED WORK

Self-training via pseudolabeling (Lee, 2013) or min-entropy objectives (Grandvalet & Bengio, 2005) has been widely used in both semi-supervised learning (Laine & Aila, 2016; Tarvainen & Valpola, 2017; Iscen et al., 2019; Yalniz et al., 2019; Berthelot et al., 2019; Xie et al., 2020; Sohn et al., 2020) and unsupervised domain adaptation (Long et al., 2013; French et al., 2017; Saito et al., 2017; Shu et al., 2018; Zou et al., 2019). Our paper studies input consistency regularization, which enforces stability of the prediction w.r.t transformations of the unlabeled data. In practice, these transformations include adversarial perturbations, which was proposed as the VAT objective (Miyato et al., 2018), as well as data augmentations (Xie et al., 2019).

For unsupervised learning, our self-training objective is closely related to BYOL (Grill et al., 2020), a recent state-of-the-art method which trains a student model to match the representations predicted by a teacher model on strongly augmented versions of the input. Contrastive learning is another popular method for unsupervised representation learning which encourages representations of "positive pairs", ideally consisting of examples from the same class, to be close, while pushing negative pairs far apart (Mikolov et al., 2013; Oord et al., 2018; Arora et al., 2019). Recent works in contrastive learning achieve state-of-the-art representation quality by using strong data augmentation to form positive pairs (Chen et al., 2020a;b). The role of data augmentation here is in spirit similar to our use of input consistency regularization. Less related to our setting are algorithms which learn representations by solving self-supervised pretext tasks, such as inpainting and predicting rotations (Pathak et al., 2016; Noroozi & Favaro, 2016; Gidaris et al., 2018). Lee et al. (2020) theoretically analyze self-supervised learning, but their analysis applies to a different class of algorithms than ours.

Prior theoretical works analyze contrastive learning by assuming access to document data distributed according to a particular topic modeling setup (Tosh et al., 2020) or pairs of independent samples within the same class (Arora et al., 2019). However, the assumptions required for these analyses do not necessarily apply to vision, where positive pairs apply different data augmentations to the same image, and are therefore strongly correlated. Other papers analyze information-theoretic properties of representation learning (Tian et al., 2020; Tsai et al., 2020).

Prior works analyze continuity or "cluster" assumptions for semi-supervised learning which are related to our notion of expansion (Seeger, 2000; Rigollet, 2007; Singh et al., 2009; Urner & Ben-David, 2013). However, these papers leverage unlabeled data using non-parametric methods, requiring unlabeled sample complexity that is exponential in the dimension. On the other hand, our analysis is for parametric methods, and therefore the unlabeled sample complexity can be low when a neural net can separate the ground-truth classes with large margin.

Co-training is a classical version of self-training which requires two distinct "views" (i.e., feature subsets) of the data, each of which can be used to predict the true label on its own (Blum & Mitchell, 1998; Dasgupta et al., 2002; Balcan et al., 2005). For example, to predict the topic of a webpage, one view could be the incoming links and another view could be the words in the page. The original co-training algorithms (Blum & Mitchell, 1998; Dasgupta et al., 2002) assume that the two views are independent conditioned on the true label and leverage this independence to obtain accurate pseudolabels for the unlabeled data. By contrast, if we cast our setting into the co-training framework by treating an example and a randomly sampled neighbor as the two views of the data, the two views are highly correlated. Balcan et al. (2005) relax the requirement on independent views of co-training, also by using an "expansion" assumption. Our assumption is closely related to theirs and conceptually equivalent if we cast our setting into the co-training framework by treating neighboring examples are two views. However, their analysis requires confident pseudolabels to all be accurate and does not rigorously account for potential propagation of errors from their algorithm. In contrast, our contribution is to propose and analyze an objective function involving input consistency regularization whose minimizer *denoises* errors from potentially incorrect pseudolabels. We also provide finite sample complexity bounds for the neural network hypothesis class and analyze unsupervised learning algorithms.

Alternative theoretical analyses of unsupervised domain adaptation assume bounded measures of discrepancy between source and target domains (Ben-David et al., 2010; Zhang et al., 2019). Balcan & Blum (2010) propose a PAC-style framework for analyzing semi-supervised learning, but their bounds require the user to specify a notion of compatability which incorporates prior knowledge about the data, and do not apply to domain adaptation. Globerson et al. (2017) demonstrate semi-supervised learning can outperform supervised learning in labeled sample complexity but assume full knowledge of the unlabeled distribution. (Mobahi et al., 2020) show that for kernel methods, self-distillation, a variant of self-training, can effectively amplify regularization. Their analysis is for kernel methods, whereas our analysis applies to deep networks under data assumptions.

## 2 PRELIMINARIES AND NOTATIONS

We let $P$ denote a distribution of unlabeled examples over input space $\mathcal{X}$. For unsupervised learning, $P$ is the only relevant distribution. For unsupervised domain adaptation, we also define a source distribution $P_{\text{src}}$ and let $G_{\text{pl}}$ denote a source classifier trained on a labeled dataset sampled from $P_{\text{src}}$.

To translate these definitions to semi-supervised learning, we set $P_{\text{src}}$ and $P$ to be the same, except $P_{\text{src}}$ gives access to labels. We analyze algorithms which only depend on $P_{\text{src}}$ through $G_{\text{pl}}$.

We consider classification and assume the data is partitioned into $K$ classes, where the class of $x \in \mathcal{X}$ is given by the ground-truth $G^{\star}(x)$ for $G^{\star} : \mathcal{X} \to [K]$. We let $P_i$ denote the class-conditional distribution of $x$ conditioned on $G^{\star}(x) = i$. We assume that each example $x$ has a unique label, so $P_i$, $P_j$ have disjoint support for $i \neq j$. Let $\widehat{P} \triangleq \{x_1, \ldots, x_n\} \subset \mathcal{X}$ denote $n$ i.i.d. unlabeled training examples from $P$. We also use $\widehat{P}$ to refer to the uniform distribution over these examples. We let $F : \mathcal{X} \to \mathbb{R}^K$ denote a learned scoring function (e.g. the continuous logits output by a neural network), and $G : \mathcal{X} \to [K]$ the discrete labels induced by $F$: $G(x) \triangleq \arg\max_i F(x)_i$ (where ties are broken lexicographically).

**Pseudolabels.** Pseudolabeling methods are a form of self-training for semi-supervised learning and domain adaptation where the source classifier $G_{\text{pl}} : \mathcal{X} \to [K]$ is used to predict pseudolabels on the unlabeled target data (Lee, 2013). These methods then train a fresh classifier to fit these pseudolabels, for example, using the standard cross entropy loss: $L_{\text{pl}}(F) \triangleq \mathbb{E}_{\widehat{P}}[\ell_{\text{cross-ent}}(F(x), G_{\text{pl}}(x))]$. Our theoretical analysis applies to a pseudolabel-based objective. Other forms of self-training include entropy minimization, which is closely related, and in certain settings, equivalent to pseudolabeling where the pseudolabels are updated every iteration (Lee, 2013; Chen et al., 2020c).

## 3   EXPANSION PROPERTY AND GUARANTEES FOR UNSUPERVISED LEARNING

In this section we will first introduce our key assumption on expansion. We then study the implications of expansion for unsupervised learning. We show that if a classifier is consistent w.r.t. input transformations and predicts each class with decent probability, the learned labels will align with ground-truth classes up to permutation of the class indices (Theorem 3.6).

### 3.1   EXPANSION PROPERTY

We introduce the notion of expansion. As our theory studies objectives which enforce stability to input transformations, we will first model allowable transformations of the input $x$ by the set $\mathcal{B}(x)$, defined below. We let $\mathcal{T}$ denote some set of transformations obtained via data augmentation, and define $\mathcal{B}(x) \triangleq \{x' : \exists T \in \mathcal{T} \text{ such that } \|x' - T(x)\| \leq r\}$ to be the set of points with distance $r$ from some data augmentation of $x$. We can think of $r$ as a value much smaller than the typical norm of $x$, so the probability $P(\mathcal{B}(x))$ is exponentially small in dimension. Our theory easily applies to other choices of $\mathcal{B}$, though we set this definition as default for simplicity. Now we define the neighborhood of $x$, denoted by $\mathcal{N}(x)$, as the set of points whose transformation sets overlap with that of $x$:

$$\mathcal{N}(x) = \{x' : \mathcal{B}(x) \cap \mathcal{B}(x') \neq \emptyset\} \tag{3.1}$$

For $S \subseteq \mathcal{X}$, we define the neighborhood of $S$ as the union of neighborhoods of its elements: $\mathcal{N}(S) \triangleq \cup_{x \in S} \mathcal{N}(x)$. We now define the expansion property of the distribution $P$, which lower bounds the neighborhood size of low probability sets and captures connectivity of the distribution in input space.

**Definition 3.1** $((a, c)$-expansion). *We say that the class-conditional distribution $P_i$ satisfies $(a, c)$-expansion if for all $V \subseteq \mathcal{X}$ with $P_i(V) \leq a$, the following holds:*

$$P_i(\mathcal{N}(V)) \geq \min\{cP_i(V), 1\} \tag{3.2}$$

*If $P_i$ satisfies $(a, c)$-expansion for all $\forall i \in [K]$, then we say $P$ satisfies $(a, c)$-expansion.*

We note that this definition considers the *population* distribution, and expansion is not expected to hold on the training set, because all empirical examples are far away from each other, and thus the neighborhoods of training examples do not overlap. The notion is closely related to the *Cheeger constant*, which is used to bound mixing times and hitting times for sampling from continuous distributions (Lovász & Vempala, 2007; Zhang et al., 2017), and *small-set expansion*, which quantifies connectivity of graphs (Hoory et al., 2006; Raghavendra & Steurer, 2010). In particular, when the neighborhood is defined to be the collection of points with $\ell_2$ distance at most $r$ from the set, then the expansion factor $c$ is bounded below by $\exp(\eta r)$, where $\eta$ is the Cheeger constant (Zhang et al., 2017). In Section E.1, we use GANs to demonstrate that expansion is a realistic property in vision. For unsupervised learning, we require expansion with $a = 1/2$ and $c > 1$:

**Assumption 3.2** (Expansion requirement for unsupervised learning). *We assume that $P$ satisfies $(1/2, c)$-expansion on $\mathcal{X}$ for $c > 1$.*

We also assume that ground-truth classes are separated in input space. We define the population consistency loss $R_\mathcal{B}(G)$ as the fraction of examples where $G$ is not robust to input transformations:

$$R_\mathcal{B}(G) \triangleq \mathbb{E}_P[\mathbf{1}(\exists x' \in \mathcal{B}(x) \text{ such that } G(x') \neq G(x))] \tag{3.3}$$

We state our assumption that ground-truth classes are far in input space below:

**Assumption 3.3** (Separation). *We assume $P$ is $\mathcal{B}$-separated with probability $1 - \mu$ by ground-truth classifier $G^\star$, as follows: $R_\mathcal{B}(G^\star) \leq \mu$.*

Our accuracy guarantees in Theorems 4.3 and 3.6 will depend on $\mu$. We expect $\mu$ to be small or negligible (e.g. inverse polynomial in dimension). The separation requirement requires the distance between two classes to be larger than $2r$, the $\ell_2$ radius in the definition of $\mathcal{B}(\cdot)$. However, $r$ can be much smaller than the norm of a typical example, so our expansion requirement can be weaker than a typical notion of "clustering" which requires intra-class distances to be smaller than inter-class distances. We demonstrate this quantitatively, starting with a mixture of Gaussians.

**Example 3.4** (Mixture of isotropic Gaussians). *Suppose $P$ is a mixture of $K$ Gaussians $P_i \triangleq \mathcal{N}(\tau_i, \frac{1}{d} I_{d \times d})$ with isotropic covariance and $K < d$, corresponding to $K$ separate classes.[1] Suppose the transformation set $\mathcal{B}(x)$ is an $\ell_2$-ball with radius $\frac{1}{2\sqrt{d}}$ around $x$, so there is no data augmentation and $r = \frac{1}{2\sqrt{d}}$. Then $P$ satisfies $(0.5, 1.5)$-expansion. Furthermore, if the minimum distance between means satisfies $\min_{i,j} \|\tau_i - \tau_j\|_2 \gtrsim \frac{\sqrt{\log d}}{\sqrt{d}}$, then $P$ is $\mathcal{B}$-separated with probability $1 - 1/\text{poly}(d)$.*

In the example above, the population distribution satisfies expansion, but the empirical distribution *does not*. The minimum distance between any two empirical examples is $\Omega(1)$ with high probability, so they cannot be neighbors of each other when $r = \frac{1}{2\sqrt{d}}$. Furthermore, the intra-class distance, which is $\Omega(1)$, is much larger than the distance between the means, which is assumed to be $\gtrsim 1/\sqrt{d}$. Therefore, trivial distanced-based clustering algorithms on empirical samples do not apply. Our unsupervised learning algorithm in Section 3.2 can approximately recover the mixture components with polynomial samples, up to $O(1/\text{poly}(d))$ error. Furthermore, this is almost information-theoretically optimal: by total variation distance, $\Omega(\frac{1}{\sqrt{d}})$ distance between the means is required to recover the mixture components.

The example extends to log-concave distributions via more general isoperimetric inequalities (Bobkov et al., 1999). Thus, our analysis applies to the setting of prior work (Chen et al., 2020c), which studied self-training with linear models on mixtures of Gaussian or log-concave distributions.

The main benefit of our analysis, however, is that it holds for much richer family of distributions than Gaussians, compared to prior work on self-training which only considered Gaussian or near-Gaussian distributions (Raghunathan et al., 2020; Chen et al., 2020c; Kumar et al., 2020). We demonstrate this in the following mixture of manifolds example:

**Example 3.5** (Mixture of manifolds). *Suppose each class-conditional distribution $P_i$ over an ambient space $\mathbb{R}^{d'}$, where $d' > d$, is generated by some $\kappa$-bi-Lipschitz[2] generator $Q_i : \mathbb{R}^d \to \mathbb{R}^{d'}$ on latent variable $z \in \mathbb{R}^d$:*

$$x \sim P_i \Leftrightarrow x = Q_i(z), z \sim \mathcal{N}(0, \frac{1}{d} \cdot I_{d \times d})$$

*We set the transformation set $\mathcal{B}(x)$ to be an $\ell_2$-ball with radius $\frac{\kappa}{2\sqrt{d}}$ around $x$, so there is no data augmentation and $r = \frac{\kappa}{2\sqrt{d}}$. Then, $P$ satisfies $(0.5, 1.5)$-expansion.*

Figure 1 (right) provides a illustration of expansion on manifolds. Note that as long as $\kappa \ll d^{1/4}$, the radius $\kappa/(2\sqrt{d})$ is much smaller than the norm of the data points (which is at least on the order of $1/\kappa$). This suggests that the generator can non-trivially scramble the space and still maintain meaningful expansion with small radius. In Section C.2, we prove the claims made in our examples.

---

[1]The classes are not disjoint, as is assumed by our theory for simplicity. However, they are approximately disjoint, and it is easy to modify our analysis to accomodate this. We provide details in Section C.2.

[2]A $\kappa$-bi-Lipschitz function $f$ satisfies that $\frac{1}{\kappa}\|x - y\| \leq |f(x) - f(y)| \leq \kappa\|x - y\|$.

## 3.2 POPULATION GUARANTEES FOR UNSUPERVISED LEARNING

We design an unsupervised learning objective which leverages the expansion and separation properties. Our objective is on the population distribution, but it is parametric, so we can extend it to the finite sample case in Section 3.3. We wish to learn a classifier $G : \mathcal{X} \to [K]$ using only unlabeled data, such that predicted classes align with ground-truth classes. Note that without observing any labels, we can only learn ground-truth classes up to permutation, leading to the following permutation-invariant error defined for a classifier $G$:

$$\text{Err}_{\text{unsup}}(G) \triangleq \min_{\text{permutation } \pi:[K] \to [K]} \mathbb{E}[\mathbf{1}(\pi(G(x)) \neq G^{\star}(x))]$$

We study the following unsupervised population objective over classifiers $G : \mathcal{X} \to [K]$, which encourages input consistency while ensuring that predicted classes have sufficient probability.

$$\min_{G} \ R_{\mathcal{B}}(G) \quad \text{subject to} \quad \min_{y \in [K]} \mathbb{E}_P[\mathbf{1}(G(x) = y)] > \max\left\{\frac{2}{c-1}, 2\right\} R_{\mathcal{B}}(G) \qquad (3.4)$$

Here $c$ is the expansion coefficient in Assumption 3.2. The constraint ensures that the probability of any predicted class is larger than the input consistency loss. Let $\rho \triangleq \min_{y \in [K]} P(\{x : G^{\star}(x) = y\})$ denote the probability of the smallest ground-truth class. The following theorem shows that when $P$ satisfies expansion and separation, the global minimizer of the objective (3.4) will have low error.

**Theorem 3.6.** *Suppose that Assumptions 3.2 and 3.3 hold for some $c, \mu$ such that $\rho > \max\{\frac{2}{c-1}, 2\}\mu$. Then any minimizer $\widehat{G}$ of (3.4) satisfies*

$$\text{Err}_{\text{unsup}}(\widehat{G}) \leq \max\left\{\frac{c}{c-1}, 2\right\}\mu \qquad (3.5)$$

In Section C, we provide the proof of Theorem 3.6 as well as a variant of the theorem which holds for a weaker additive notion of expansion. By applying the generalization bounds of Section 3.3, we can convert Theorem 3.6 into a finite-sample guarantees that are polynomial in margin and Lipschitzness of the model (see Theorem D.1).

Our objective is reminiscent of recent methods which achieve state-of-the-art results in unsupervised representation learning: SimCLR (Chen et al., 2020a), MoCov2 (He et al., 2020; Chen et al., 2020b), and BYOL (Grill et al., 2020). Unlike our algorithm, these methods do not predict discrete labels, but rather, directly predict a representation which is consistent under input transformations, However, our analysis still suggests an explanation for why input consistency regularization is so vital for these methods: assuming the data satisfies expansion, it encourages representations to be similar over the *entire* class, so the representations will capture ground-truth class structure.

Chen et al. (2020a) also observe that using more aggressive data augmentation for regularizing input stability results in significant improvements in representation quality. We remark that our theory offers a potential explanation: in our framework, strengthening augmentation increases the size of the neighborhood, resulting in a larger expansion factor $c$ and improving the accuracy bound (3.5).

## 3.3 FINITE SAMPLE GUARANTEES FOR DEEP LEARNING MODELS

In this section, we show that if the ground-truth classes are separable by a neural net with large robust margin, then generalization can be good. The main advantage of Theorem 3.6 and Theorem 4.3 over prior work is that they analyze parametric objectives, so finite sample guarantees immediately hold via off-the-shelf generalization bounds. Prior work on continuity or "cluster" assumptions related to expansion require nonparametric techniques with a sample complexity that is exponential in dimension (Seeger, 2000; Rigollet, 2007; Singh et al., 2009; Urner & Ben-David, 2013).

We apply the generalization bound of (Wei & Ma, 2019b) based on a notion of all-layer margin, though any other bound would work. The all-layer margin measures the stability of the neural net to simultaneous perturbations to each hidden layer. Formally, suppose that $G(x) \triangleq \arg\max_i F(x)_i$ is the prediction of some feedforward neural network $F : \mathcal{X} \to \mathbb{R}^K$ which computes the following function: $F(x) = W_p\phi(\cdots\phi(W_1 x)\cdots)$ with weight matrices $\{W_i\}_{i=1}^p$. Let $q$ denote the maximum dimension of any hidden layer. Let $m(F, x, y) \geq 0$ denote the all-layer margin at example $x$ for

label $y$, defined formally in Section D.2. For now, we simply note that $m$ has the property that if $G(x) \neq y$, then $m(F, x, y) = 0$, so we can upper bound the 0-1 loss by thresholding the all-layer margin: $\mathbf{1}(G(x) \neq y) \leq \mathbf{1}(m(F, x, y) \geq t)$ for any $t > 0$. We can also define a variant that measures robustness to input transformations: $m_{\mathcal{B}}(F, x) \triangleq \min_{x' \in \mathcal{B}(x)} m\left(F, x', \arg\max_i F(x)_i\right)$. The following result states that large all-layer margin implies good generalization for the input consistency loss, which appears in the objective (3.4).

**Theorem 3.7** (Extension of Theorem 3.1 of (Wei & Ma, 2019b)). *With probability $1 - \delta$ over the draw of the training set $\widehat{P}$, all neural networks $G = \arg\max_i F_i$ of the form $F(x) \triangleq W_p \phi(\cdots \phi(W_1 x))$ will satisfy*

$$R_{\mathcal{B}}(G) \leq \mathbb{E}_{\widehat{P}}[\mathbf{1}(m_{\mathcal{B}}(F, x) \leq t)] + \widetilde{O}\left(\frac{\sum_i \sqrt{q} \|W_i\|_F}{t\sqrt{n}}\right) + \zeta \qquad (3.6)$$

*for all choices of $t > 0$, where $\zeta \triangleq O\left(\sqrt{(\log(1/\delta) + p \log n)/n}\right)$ is a low-order term, and $\widetilde{O}(\cdot)$ hides poly-logarithmic factors in $n$ and $d$.*

A similar bound can be expressed for other quantities in (3.4), and is provided in Section D.2. In Section D.1, we plug our bounds into Theorem 3.6 and Theorem 4.3 to provide accuracy guarantees which depend on the unlabeled training set. We provide a proof overview in Section D.2, and in Section D.3, we provide a data-dependent lower bound on the all-layer margin that scales inversely with the Lipschitzness of the model, measured via the Jacobian and hidden layer norms on the training data. These quantities have been shown to be typically well-behaved (Arora et al., 2018; Nagarajan & Kolter, 2019; Wei & Ma, 2019a). In Section E.2, we empirically show that explicitly regularizing the all-layer margin improves the performance of self-training.

## 4 DENOISING PSEUDOLABELS FOR SEMI-SUPERVISED LEARNING AND DOMAIN ADAPTATION

We study semi-supervised learning and unsupervised domain adaptation settings where we have access to unlabeled data and a pseudolabeler $G_{\text{pl}}$. This setting requires a more complicated analysis than the unsupervised learning setting because pseudolabels may be inaccurate, and a student classifier could amplify these mistakes. We design a population objective which measures input transformation consistency and pseudolabel accuracy. Assuming expansion and separation, we show that the minimizer of this objective will have high accuracy on *ground-truth* labels.

We assume access to pseudolabeler $G_{\text{pl}}(\cdot)$, obtained via training a classifier on the labeled source data in the domain adaptation setting or on the labeled data in the semi-supervised setting. With access to pseudolabels, we can aim to recover the true labels exactly, rather than up to permutation as in Section 3.2. For $G, G' : \mathcal{X} \to [K]$, define $L_{0\text{-}1}(G, G') \triangleq \mathbb{E}_P[\mathbf{1}(G(x) \neq G'(x))]$ to be the disagreement between $G$ and $G'$. The error metric is the standard 0-1 loss on ground-truth labels: $\text{Err}(G) \triangleq L_{0\text{-}1}(G, G^\star)$. Let $\mathcal{M}(G_{\text{pl}}) \triangleq \{x : G_{\text{pl}}(x) \neq G^\star(x)\}$ denote the set of mistakenly pseudolabeled examples. We require the following assumption on expansion, which intuitively states that each subset of $\mathcal{M}(G_{\text{pl}})$ has a large enough neighborhood.

**Assumption 4.1** ($P$ expands on sets smaller than $\mathcal{M}(G_{\text{pl}})$). *Define $\bar{a} \triangleq \max_i \{P_i(\mathcal{M}(G_{\text{pl}}))\}$ to be the maximum fraction of incorrectly pseudolabeled examples in any class. We assume that $\bar{a} < 1/3$ and $P$ satisfies $(\bar{a}, \bar{c})$-expansion for $\bar{c} > 3$. We express our bounds in terms of $c \triangleq \min\{1/\bar{a}, \bar{c}\}$.*

Note that the above requirement $c > 3$ is more demanding than the condition $c > 1$ required in the unsupervised learning setting (Assumption 3.2). The larger $c > 3$ accounts for the possibility that mistakes in the pseudolabels can adversely affect the learned classifier in a worst-case manner. This concern doesn't apply to unsupervised learning because pseudolabels are not used. For the toy distributions in Examples 3.4 and 3.5, we can increase the radius of the neighborhood by a factor of 3 to obtain (0.16, 6)-expansion, which is enough to satisfy the requirement in Assumption 4.1.

On the other hand, Assumption 4.1 is less strict than Assumption 3.2 in the sense that expansion is only required for small sets with mass less than $\bar{a}$, the pseudolabeler's worst-case error on a class, which can be much smaller than $a = 1/2$ required in Assumption 3.2. Furthermore, the

unsupervised objective (3.4) has the constraint that the input consistency regularizer is not too large, whereas no such constraint is necessary for this setting. We remark that Assumption 4.1 can also be relaxed to directly consider expansion of subsets of incorrectly pseudolabeled examples, also with a looser requirement on the expansion factor $c$ (see Section B.1). We design the following objective over classifiers $G$, which fits the classifier to the pseudolabels while regularizing input consistency:

$$\min_G \mathcal{L}(G) \triangleq \frac{c+1}{c-1} L_{\text{0-1}}(G, G_{\text{pl}}) + \frac{2c}{c-1} R_{\mathcal{B}}(G) - \text{Err}(G_{\text{pl}}) \tag{4.1}$$

The objective optimizes weighted combinations of $R_{\mathcal{B}}(G)$, the input consistency regularizer, and $L_{\text{0-1}}(G, G_{\text{pl}})$, the loss for fitting pseudolabels, and is related to recent successful algorithms for semi-supervised learning (Sohn et al., 2020; Xie et al., 2020). We can show that $\mathcal{L}(G) \geq 0$ always holds. The following lemma bounds the error of $G$ in terms of the objective value.

**Lemma 4.2.** *Suppose Assumption 4.1 holds. Then the error of classifier $G : \mathcal{X} \to [K]$ is bounded in terms of consistency w.r.t. input transformations and accuracy on pseudolabels:* $\text{Err}(G) \leq \mathcal{L}(G)$.

When expansion and separation both hold, we show that minimizing (4.1) leads to a classifier that can *denoise* the pseudolabels and improve on their ground-truth accuracy.

**Theorem 4.3.** *Suppose Assumptions 4.1 and 3.3 hold. Then for any minimizer $\widehat{G}$ of (4.1), we have*

$$\text{Err}(\widehat{G}) \leq \frac{2}{c-1} \text{Err}(G_{\text{pl}}) + \frac{2c}{c-1} \mu \tag{4.2}$$

We provide a proof sketch in Section A, and the full proof in Section B.1. Our result explains the perhaps surprising fact that self-training with pseudolabeling often improves over the pseudolabeler even though no additional information about true labels is provided. In Theorem D.2, we translate Theorem 4.3 into a finite-sample guarantee by using the generalization bounds in Section 3.3.

At a first glance, the error bound in Theorem 4.3 appears weaker than Theorem 3.6 because of the additional dependence on $\text{Err}(G_{\text{pl}})$. This discrepancy is due to weaker requirements on the expansion and the value of the input consistency regularizer. First, Section 3.2 requires expansion on all sets with probability less than $1/2$, whereas Assumption 4.1 only requires expansion on sets with probability less than $\bar{a}$, which can be much smaller than $1/2$. Second, the error bounds in Section 3.2 only apply to classifiers with small values of $R_{\mathcal{B}}(G)$, as seen in (3.4). On the other hand, Lemma 4.2 gives an error bound for *all* classifiers, regardless of $R_{\mathcal{B}}(G)$. Indeed, strengthening the expansion requirement to that of Section 3.2 would allow us to obtain accuracy guarantees similar to Theorem 3.6 for pseudolabel-trained classifiers with low input consistency regularizer value.

**Experiments** In Section E.1, we provide details for the GAN experiment in Figure 1. We also provide empirical evidence for our theoretical intuition that self-training with input consistency regularization succeeds because the algorithm denoises incorrectly pseudolabeled examples with correctly pseudolabeled neighbors (Figure 3). In Section E.2, we perform ablation studies for pseudolabeling which show that components of our theoretical objective (4.1) do improve performance.

## 5 CONCLUSION

In this work, we propose an expansion assumption on the data which allows for a unified theoretical analysis of self-training for semi-supervised and unsupervised learning. Our assumption is realistic for real-world datasets, particularly in vision. Our analysis is applicable to deep neural networks and can explain why algorithms based on self-training and input consistency regularization can perform so well on unlabeled data. We hope that this assumption can facilitate future theoretical analyses and inspire theoretically-principled algorithms for semi-supervised and unsupervised learning. For example, an interesting question for future work is to extend our assumptions to analyze domain adaptation algorithms based on aligning the source and target (Hoffman et al., 2018).

## ACKNOWLEDGEMENTS

We would like to thank Ananya Kumar for helpful comments and discussions. CW acknowledges support from a NSF Graduate Research Fellowship. TM is also partially supported by the Google Faculty Award, Stanford Data Science Initiative, and the Stanford Artificial Intelligence Laboratory. The authors would also like to thank the Stanford Graduate Fellowship program for funding.

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

# A    PROOF SKETCH FOR THEOREM 4.3

We provide a proof sketch for Lemma 4.2 for the extreme case where the input consistency regularizer is 0 for all examples, i.e. $G(x) = G(x') \; \forall x \in \mathcal{X}, x' \in \mathcal{B}(x)$, so $R_\mathcal{B}(G) = 0$. For this proof sketch, we also make an additional restriction to the case when $L_{\text{0-1}}(G, G_{\text{pl}}) = \text{Err}(G_{\text{pl}})$.

We first introduce some general notation. For sets $U, V \subseteq \mathcal{X}$, we use $U \setminus V$ to denote $\{x : x \in U, x \notin V\}$, and $\cap, \cup$ denote set intersection and union, respectively. Let $\overline{U} \triangleq \mathcal{X} \setminus U$ denote the complement of $U$.

Let $\mathcal{C}_i \triangleq \{x : G^\star(x) = i\}$ denote the set of examples with ground-truth label $i$. For $S \subseteq \mathcal{X}$, we define $\mathcal{N}^\star(S)$ to be the neighborhood of $S$ with neighbors restricted to the same class: $\mathcal{N}^\star(S) \triangleq \cup_{i \in [K]} (\mathcal{N}(S \cap \mathcal{C}_i) \cap \mathcal{C}_i)$. The following key claims will consider two sets: the set of correctly pseudolabeled examples on which the classifier makes mistakes, $\{x : G(x) \neq G_{\text{pl}}(x) \text{ and } x \notin \mathcal{M}(G_{\text{pl}})\}$, and the set of examples where both classifier and pseudolabeler disagree with the ground truth, $\mathcal{M}(F) \cap \mathcal{M}(G_{\text{pl}})$. The claims below use the expansion property to show that

$$P(\{x : G(x) \neq G_{\text{pl}}(x) \text{ and } x \notin \mathcal{M}(G_{\text{pl}})\}) > P(\mathcal{M}(F) \cap \mathcal{M}(G_{\text{pl}}))$$

**Claim A.1.** *In the setting of Theorem 4.3, define the set $V \triangleq \mathcal{M}(G) \cap \mathcal{M}(G_{\text{pl}})$. Define $q \triangleq \frac{\text{Err}(G_{\text{pl}})}{c-1}$. By expansion (Assumption 4.1), if $P(V) > q$, then $P(\mathcal{N}^\star(V) \setminus \mathcal{M}(G_{\text{pl}})) > P(V)$.*

A more general version of Claim A.1 is given by Lemma B.7 in Section B.2. For a visualization of $V$ and $\mathcal{N}^\star(V) \setminus \mathcal{M}(G_{\text{pl}})$, refer to Figure 2.

**Claim A.2.** *Suppose the input consistency regularizer is 0 for all examples, i.e., $\forall x \in \mathcal{X}, x' \in \mathcal{B}(x)$, it holds that $G(x) = G(x')$. Then it follows that*

$$\{x : G(x) \neq G_{\text{pl}}(x) \text{ and } x \notin \mathcal{M}(G_{\text{pl}})\} \supseteq \mathcal{N}^\star(V) \setminus \mathcal{M}(G_{\text{pl}})$$

Figure 2 outlines the proof of this claim. Claim B.5 in Section B provides a more general version of Claim A.2 in the case where $R_\mathcal{B}(G) > 0$. Given the above, the proof of Lemma 4.2 follows by a counting argument.

*Proof sketch of Lemma 4.2 for simplified setting.* Assume for the sake of contradiction that $P(V) > q$. We can decompose the errors of $G$ on the pseudolabels as follows:

$$L_{\text{0-1}}(G, G_{\text{pl}}) \geq \mathbb{E}[\mathbf{1}(G(x) \neq G_{\text{pl}}(x) \text{ and } x \notin \mathcal{M}(G_{\text{pl}}))] + \mathbb{E}[\mathbf{1}(G(x) \neq G_{\text{pl}}(x) \text{ and } x \in \mathcal{M}(G_{\text{pl}}))]$$

We lower bound the first term by $P(V)$ by Claims A.1 and A.2. For the latter term, we note that if $x \in \mathcal{M}(G_{\text{pl}}) \setminus V$, then $G(x) = G^\star(x) \neq G_{\text{pl}}(x)$. Thus, the latter term has lower bound $P(\mathcal{M}(G_{\text{pl}})) - P(V)$. As a result, we obtain

$$L_{\text{0-1}}(G, G_{\text{pl}}) > P(V) + P(\mathcal{M}(G_{\text{pl}})) - P(V) = \text{Err}(G_{\text{pl}})$$

which contradicts our simplifying assumption that $L_{\text{0-1}}(G, G_{\text{pl}}) = \text{Err}(G_{\text{pl}})$. Thus, $G$ disagrees with $G^\star$ at most $q$ fraction of examples in $\mathcal{M}(G_{\text{pl}})$. To complete the proof, we note that $G$ also disagrees with $G^\star$ on at most $q$ fraction of examples outside of $\mathcal{M}(G_{\text{pl}})$, or else $L_{\text{0-1}}(G, G_{\text{pl}})$ would again be too high. $\qquad\square$

# B    PROOFS FOR DENOISING PSEUDOLABELS

In this section, we will provide the proof of Theorem 4.3. Our analysis will actually rely on a weaker *additive* notion of expansion, defined below. We show that the multiplicative definition in Definition 3.1 will imply that the additive variant holds.

## B.1    RELAXATION OF EXPANSION ASSUMPTION FOR PSEUDOLABELING

In this section, we provide a proof of a relaxed version of Theorem 4.3. We will then reduce Theorem 4.3 to this relaxed version in Section B.2. It will be helpful to restrict the notion of neighborhood

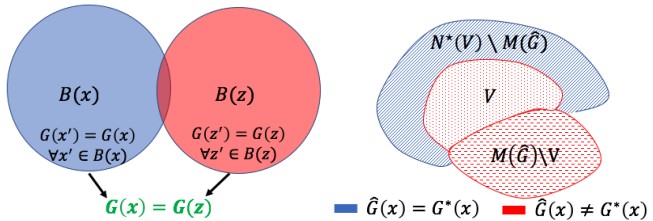

Figure 2: To prove Claim A.2, we first note that in the simplified setting, if $\mathcal{B}(x) \cap \mathcal{B}(z) \neq \emptyset$ then $G(x) = G(z)$ by the assumption that $R_{\mathcal{B}}(G) = 0$ (see **left**). By the definition of $\mathcal{N}^\star(\cdot)$, this implies that all points $x \in \mathcal{N}^\star(V) \setminus \mathcal{M}(G_{\text{pl}})$ must satisfy $G(x) \neq G^\star(x)$, as $x$ matches the label of its neighbor in $V \subseteq \mathcal{M}(G)$. However, all points in $\mathcal{X} \setminus \mathcal{M}(G_{\text{pl}})$ must satisfy $G_{\text{pl}}(x) = G^\star(x)$, and therefore $G(x) \neq G_{\text{pl}}(x)$. These sets are depicted on the **right**.

to only examples in the same ground-truth class: define $\mathcal{N}^\star(x) \triangleq \{x' : x' \in \mathcal{N}(x) \text{ and } G^\star(x') = G^\star(x)\}$ and $\mathcal{N}^\star(S) \triangleq \cup_{x \in S} \mathcal{N}^\star(x)$. Note that the following relation between $\mathcal{N}(S)$ and $\mathcal{N}^\star(S)$ holds in general:

$$\mathcal{N}^\star(S) = \cup_{i \in [K]} \left( \mathcal{N}(S \cap \mathcal{C}_i) \cap \mathcal{C}_i \right)$$

We will define the additive notion of expansion on subsets of $\mathcal{X}$ below.

**Definition B.1** ($(q, \alpha)$-additive-expansion on a set $S$). *We say that $P$ satisfies $(q, \alpha)$-additive-expansion on $S \subseteq \mathcal{X}$ if for all $V \subseteq S$ with $P(V) > q$, the following holds:*

$$P(\mathcal{N}^\star(V) \setminus S) = \sum_{i \in [K]} P(\mathcal{N}(V \cap \mathcal{C}_i) \cap \mathcal{C}_i \setminus S) > P(V) + \alpha$$

In other words, any sufficiently large subset of $S$ must have a sufficiently large neighborhood of examples sharing the same ground-truth label. For the remainder of this section, we will analyze this additive notion of expansion. In Section B.2, we will reduce multiplicative expansion (Definition 3.1) to our additive definition above.

Now for a given classifier, define the robust set of $G$, $\mathcal{S}_{\mathcal{B}}(G)$, to be the set of inputs for which $G$ is robust under $\mathcal{B}$-transformations:

$$\mathcal{S}_{\mathcal{B}}(G) = \{x : G(x) = G(x') \; \forall x' \in \mathcal{B}(x)\}$$

The following theorem shows that if the classifier $G$ is $\mathcal{B}$-robust and fits the pseudolabels sufficiently well, classification accuracy on *true* labels will be good.

**Theorem B.2.** *For a given pseudolabeler $G_{\text{pl}} : \mathcal{X} \to \{1, \ldots, K\}$, suppose that $P$ has $(q, \alpha)$-additive-expansion on $\mathcal{M}(G_{\text{pl}})$ for some $q, \alpha$. Suppose that $G$ fits the pseudolabels with sufficient accuracy and robustness:*

$$\mathbb{E}_P[\mathbf{1}(G(x) \neq G_{\text{pl}}(x) \text{ or } x \notin \mathcal{S}_{\mathcal{B}}(G))] \leq \text{Err}(G_{\text{pl}}) + \alpha \tag{B.1}$$

*Then $G$ satisfies the following error bound:*

$$\text{Err}(G) \leq 2(q + R_{\mathcal{B}}(G)) + \mathbb{E}_P[\mathbf{1}(G(x) \neq G_{\text{pl}}(x))] - \text{Err}(G_{\text{pl}})$$

To interpret this statement, suppose $G$ fits the pseudolabels with error rate at most $\text{Err}(G_{\text{pl}})$ and (B.1) holds. Then $\text{Err}(G) \leq 2(q + R_{\mathcal{B}}(G))$, so if $G$ is robust to $\mathcal{B}$-perturbations on the *population* distribution, the accuracy of $G$ is high.

Towards proving Theorem B.2, we consider three disjoint subsets of $\mathcal{M}(G) \cap \mathcal{S}_{\mathcal{B}}(G)$:

$$\mathcal{M}_1 \triangleq \{x : G(x) = G_{\text{pl}}(x), G_{\text{pl}}(x) \neq G^\star(x), \text{ and } x \in \mathcal{S}_{\mathcal{B}}(G)\}$$

$$\mathcal{M}_2 \triangleq \{x : G(x) \neq G_{\text{pl}}(x), G_{\text{pl}}(x) \neq G^\star(x), G(x) \neq G^\star(x), \text{ and } x \in \mathcal{S}_{\mathcal{B}}(G)\}$$

$$\mathcal{M}_3 \triangleq \{x : G(x) \neq G_{\text{pl}}(x), G_{\text{pl}}(x) = G^\star(x), \text{ and } x \in \mathcal{S}_{\mathcal{B}}(G)\}$$

We first bound the probability of $\mathcal{M}_1 \cup \mathcal{M}_2$.

**Lemma B.3.** *In the setting of Theorem B.2, we have $P(\mathcal{S}_\mathcal{B}(G) \cap \mathcal{M}(G_{\mathrm{pl}}) \cap \mathcal{M}(G)) \leq q$. As a result, since it holds that $\mathcal{M}_1 \cup \mathcal{M}_2 \subseteq \mathcal{S}_\mathcal{B}(G) \cap \mathcal{M}(G_{\mathrm{pl}}) \cap \mathcal{M}(G)$, it immediately follows that $P(\mathcal{M}_1 \cup \mathcal{M}_2) \leq q$.*

The proof relies on the following claims.

**Claim B.4.** *In the setting of Theorem 4.3, define $U \triangleq \mathcal{N}^\star(\mathcal{S}_\mathcal{B}(G) \cap \mathcal{M}(G_{\mathrm{pl}}) \cap \mathcal{M}(G)) \setminus \mathcal{M}(G_{\mathrm{pl}})$. For any $x \in U \cap \mathcal{S}_\mathcal{B}(G)$, it holds that $G_{\mathrm{pl}}(x) \neq G(x)$ and $G(x) \neq G^\star(x)$.*

*Proof.* For any $x \in U \subseteq \mathcal{N}^\star(\mathcal{S}_\mathcal{B}(G) \cap \mathcal{M}(G_{\mathrm{pl}}) \cap \mathcal{M}(G))$, there exists $x' \in \mathcal{S}_\mathcal{B}(G) \cap \mathcal{M}(G_{\mathrm{pl}}) \cap \mathcal{M}(G)$ such that $\mathcal{B}(x) \cap \mathcal{B}(x') \neq \emptyset$ and $G^\star(x) = G^\star(x')$ by definition of $\mathcal{N}^\star(\cdot)$. Choose $z \in \mathcal{B}(x) \cap \mathcal{B}(x')$. As $x, x' \in \mathcal{S}_\mathcal{B}(G)$, by definition of $\mathcal{S}_\mathcal{B}(G)$ we also must have $G(x) = G(z) = G(x')$. Furthermore, as $x' \in \mathcal{M}(G)$, $G(x') \neq G^\star(x')$. Since $G^\star(x) = G^\star(x')$, it follows that $G(x) \neq G^\star(x)$.

As $U \cap \mathcal{M}(G_{\mathrm{pl}}) = \emptyset$ by definition of $U$, $G_{\mathrm{pl}}$ much match the ground-truth classifier on $U$, so $G_{\mathrm{pl}}(x) = G^\star(x)$. It follows that $G(x) \neq G_{\mathrm{pl}}(x)$, as desired. $\square$

**Claim B.5.** *In the setting of Theorem B.2, define $U \triangleq \mathcal{N}^\star(\mathcal{S}_\mathcal{B}(G) \cap \mathcal{M}(G_{\mathrm{pl}}) \cap \mathcal{M}(G)) \setminus \mathcal{M}(G_{\mathrm{pl}})$. If $P(\mathcal{S}_\mathcal{B}(G) \cap \mathcal{M}(G_{\mathrm{pl}}) \cap \mathcal{M}(G)) > q$, then*

$$P(U \cap \mathcal{S}_\mathcal{B}(G)) > P(\mathcal{M}(G_{\mathrm{pl}})) + P(\mathcal{S}_\mathcal{B}(G)) + \alpha - 1 - P(\mathcal{S}_\mathcal{B}(G) \cap \mathcal{M}(G_{\mathrm{pl}}) \cap \overline{\mathcal{M}(G)})$$

*Proof.* Define $V \triangleq \mathcal{S}_\mathcal{B}(G) \cap \mathcal{M}(G_{\mathrm{pl}}) \cap \mathcal{M}(G)$. By the assumption that $\mathcal{M}(G_{\mathrm{pl}})$ satifies $(q, \alpha)$-additive-expansion, if $P(V) > q$ holds, it follows that $P(U) > P(V) + \alpha$. Furthermore, we have $U \setminus \mathcal{S}_\mathcal{B}(G) \subseteq \overline{\mathcal{S}_\mathcal{B}(G) \cup \mathcal{M}(G_{\mathrm{pl}})}$ by definition of $U$ and $V$ as $U \cap \mathcal{M}(G_{\mathrm{pl}}) = \emptyset$, and so $P(U \setminus \mathcal{S}_\mathcal{B}(G)) \leq 1 - P(\mathcal{S}_\mathcal{B}(G) \cup \mathcal{M}(G_{\mathrm{pl}}))$. Thus, we obtain

$$\begin{aligned}
P(U \cap \mathcal{S}_\mathcal{B}(G)) &= P(U) - P(U \setminus \mathcal{S}_\mathcal{B}(G)) \\
&> P(V) + \alpha - 1 + P(\mathcal{S}_\mathcal{B}(G) \cup \mathcal{M}(G_{\mathrm{pl}}))
\end{aligned}$$

Now we use the principle of inclusion-exclusion to compute

$$P(\mathcal{S}_\mathcal{B}(G) \cup \mathcal{M}(G_{\mathrm{pl}})) = P(\mathcal{M}(G_{\mathrm{pl}})) + P(\mathcal{S}_\mathcal{B}(G)) - P(\mathcal{S}_\mathcal{B}(G) \cap \mathcal{M}(G_{\mathrm{pl}}))$$

Plugging into the previous, we obtain

$$\begin{aligned}
P(U \cap \mathcal{S}_\mathcal{B}(G)) &> P(\mathcal{M}(G_{\mathrm{pl}})) + P(\mathcal{S}_\mathcal{B}(G)) + \alpha - 1 + P(V) - P(\mathcal{S}_\mathcal{B}(G) \cap \mathcal{M}(G_{\mathrm{pl}})) \\
&= P(\mathcal{M}(G_{\mathrm{pl}})) + P(\mathcal{S}_\mathcal{B}(G)) + \alpha - 1 - P(\mathcal{S}_\mathcal{B}(G) \cap \mathcal{M}(G_{\mathrm{pl}}) \cap \overline{\mathcal{M}(G)})
\end{aligned}$$

where we obtained the last line because $V = \mathcal{S}_\mathcal{B}(G) \cap \mathcal{M}(G_{\mathrm{pl}}) \cap \mathcal{M}(G) \subseteq \mathcal{S}_\mathcal{B}(G) \cap \mathcal{M}(G_{\mathrm{pl}})$. $\square$

*Proof of Lemma B.3.* To complete the proof of Lemma B.3, we first compose $\mathcal{S}_\mathcal{B}(G)$ into three disjoint sets:

$$\begin{aligned}
S_1 &\triangleq \{x : G(x) = G_{\mathrm{pl}}(x)\} \cap \mathcal{S}_\mathcal{B}(G) \\
S_2 &\triangleq \{x : G(x) \neq G_{\mathrm{pl}}(x)\} \cap \mathcal{M}(G_{\mathrm{pl}}) \cap \mathcal{S}_\mathcal{B}(G) \\
S_3 &\triangleq \{x : G(x) \neq G_{\mathrm{pl}}(x)\} \cap \overline{\mathcal{M}(G_{\mathrm{pl}})} \cap \mathcal{S}_\mathcal{B}(G)
\end{aligned}$$

First, by Claim B.4 and definition of $U$, we have $\forall x \in U \cap \mathcal{S}_\mathcal{B}(G)$, $G(x) \neq G_{\mathrm{pl}}(x)$ and $x \notin \mathcal{M}(G_{\mathrm{pl}})$. Thus, it follows that $U \cap \mathcal{S}_\mathcal{B}(G) \subseteq S_3$.

Next, we claim that $V' \triangleq \mathcal{M}(G_{\mathrm{pl}}) \cap \overline{\mathcal{M}(G)} \cap \mathcal{S}_\mathcal{B}(G) \subseteq S_2$. To see this, note that for $x \in V'$, $G(x) = G^\star(x)$ and $G_{\mathrm{pl}}(x) \neq G^\star(x)$. Thus, $G(x) \neq G_{\mathrm{pl}}(x)$, and $x \in \mathcal{S}_\mathcal{B}(G) \cap \mathcal{M}(G_{\mathrm{pl}})$, which implies $x \in S_2$.

Assume for the sake of contradiction that $P(\mathcal{S}_\mathcal{B}(G) \cap \mathcal{M}(G_{\mathrm{pl}}) \cap \mathcal{M}(G)) > q$. Now we have

$$\begin{aligned}
P(\mathcal{S}_\mathcal{B}(G)) &\geq P(S_1) + P(S_2) + P(S_3) \\
&\geq P(S_1) + P(\mathcal{S}_\mathcal{B}(G) \cap \mathcal{M}(G_{\mathrm{pl}}) \cap \overline{\mathcal{M}(G)}) + P(U \cap \mathcal{S}_\mathcal{B}(G)) \\
&> P(S_1) + P(\mathcal{M}(G_{\mathrm{pl}})) + P(\mathcal{S}_\mathcal{B}(G)) + \alpha - 1 \qquad \text{(by Claim B.5)}
\end{aligned}$$

However, we also have

$$P(S_1) = 1 - \mathbb{E}_P[\mathbf{1}(G(x) \neq G_{\mathrm{pl}}(x) \text{ or } x \notin \mathcal{S}_{\mathcal{B}}(G))]$$
$$\geq 1 - \mathrm{Err}(G_{\mathrm{pl}}) - \alpha \qquad \text{(by the condition in (B.1))}$$

Plugging this in gives us $P(S_1) + P(S_2) + P(S_3) > P(\mathcal{S}_{\mathcal{B}}(G))$, a contradiction. Thus, $P(\mathcal{S}_{\mathcal{B}}(G) \cap \mathcal{M}(G_{\mathrm{pl}}) \cap \mathcal{M}(G)) \leq q$, as desired. $\qquad \square$

The next lemma bounds $P(\mathcal{M}_3)$.

**Lemma B.6.** *In the setting of Theorem B.2, the following bound holds:*

$$P(\mathcal{M}_3) \leq q + R_{\mathcal{B}}(G) + \mathbb{E}_P[\mathbf{1}(G(x) \neq G_{\mathrm{pl}}(x))] - \mathrm{Err}(G_{\mathrm{pl}})$$

*Proof.* The proof will follow from basic manipulation. First, we note that

$$\mathcal{M}_3 \cup \{x : G(x) = G_{\mathrm{pl}}(x) \text{ and } x \in \mathcal{S}_{\mathcal{B}}(G)\} \qquad (\text{B.2})$$
$$= \big(\{x : G(x) \neq G_{\mathrm{pl}}(x), G_{\mathrm{pl}}(x) = G^{\star}(x)\} \cup \{x : G(x) = G_{\mathrm{pl}}(x), G_{\mathrm{pl}}(x) = G^{\star}(x)\}$$
$$\cup \{x : G(x) = G_{\mathrm{pl}}(x), G_{\mathrm{pl}}(x) \neq G^{\star}(x)\}\big) \cap \mathcal{S}_{\mathcal{B}}(G)$$
$$= \mathcal{M}_1 \cup \{x : G_{\mathrm{pl}}(x) = G^{\star}(x) \text{ and } x \in \mathcal{S}_{\mathcal{B}}(G)\} \qquad (\text{B.3})$$

As (B.2) and (B.3) pertain to unions of disjoint sets, it follows that

$$P(\mathcal{M}_3) + P(\{x : G(x) =$$
$$G_{\mathrm{pl}}(x) \text{ and } x \in \mathcal{S}_{\mathcal{B}}(G)\}) = P(\mathcal{M}_1) + P(\{x : G_{\mathrm{pl}}(x) = G^{\star}(x) \text{ and } x \in \mathcal{S}_{\mathcal{B}}(G)\})$$

Thus, rearranging we obtain

$$P(\mathcal{M}_3) = P(\mathcal{M}_1) + P(\{x : G_{\mathrm{pl}}(x) = G^{\star}(x)\} \cap \mathcal{S}_{\mathcal{B}}(G)\})$$
$$- P(\{x : G(x) = G_{\mathrm{pl}}(x)\} \cap \mathcal{S}_{\mathcal{B}}(G)\})$$
$$\leq P(\mathcal{M}_1) + P(\{x : G_{\mathrm{pl}}(x) = G^{\star}(x)\}) - P(\{x : G(x) = G_{\mathrm{pl}}(x)\} \cap \mathcal{S}_{\mathcal{B}}(G)\})$$
$$\leq P(\mathcal{M}_1) + P(\{x : G_{\mathrm{pl}}(x) = G^{\star}(x)\}) - P(\{x : G(x) = G_{\mathrm{pl}}(x)\})$$
$$+ P(\{x : G(x) = G_{\mathrm{pl}}(x)\} \cap \overline{\mathcal{S}_{\mathcal{B}}(G)})$$
$$\leq P(\mathcal{M}_1) + P(\{x : G(x) \neq G_{\mathrm{pl}}(x)\}) - P(\mathcal{M}(G_{\mathrm{pl}})) + 1 - P(\mathcal{S}_{\mathcal{B}}(G))$$
$$= P(\mathcal{M}_1) + R_{\mathcal{B}}(G) + \mathbb{E}_P[\mathbf{1}(G(x) \neq G_{\mathrm{pl}}(x))] - \mathrm{Err}(G_{\mathrm{pl}})$$

Substituting $P(\mathcal{M}_1) \leq q$ from Lemma B.3 gives the desired result. $\qquad \square$

*Proof of Theorem B.2.* To complete the proof, we compute

$$\mathrm{Err}(G) = P(\mathcal{M}(G)) \leq P(\mathcal{M}(G) \cap \mathcal{S}_{\mathcal{B}}(G)) + P(\overline{\mathcal{S}_{\mathcal{B}}(G)})$$
$$= P(\mathcal{M}_1) + P(\mathcal{M}_2) + P(\mathcal{M}_3) + R_{\mathcal{B}}(G)$$
$$\leq 2(q + R_{\mathcal{B}}(G)) + \mathbb{E}_P[\mathbf{1}(G(x) \neq G_{\mathrm{pl}}(x))] - \mathrm{Err}(G_{\mathrm{pl}})$$
$$\text{(by Lemmas B.3 and B.6)}$$

$\qquad \square$

## B.2  PROOF OF THEOREM 4.3

In this section, we complete the proof of Theorem 4.3 by reducing Lemma 4.2 to Theorem B.2. This requires converting multiplicative expansion to $(q, \alpha)$-additive-expansion, which is done in the following lemma. Let $\mathcal{M}_i(G_{\mathrm{pl}}) \triangleq \mathcal{M}(G_{\mathrm{pl}}) \cap \mathcal{C}_i$ denote the incorrectly pseudolabeled examples with ground-truth class $i$.

**Lemma B.7.** *In the setting of Theorem 4.3, suppose that Assumption 4.1 holds. Then for any $\beta \in (0, c-1]$, $P_i$ has $(q, \alpha)$-additive-expansion on $\mathcal{M}_i(G_{\mathrm{pl}})$ for the following choice of $q, \alpha$:*

$$q = \frac{\beta P(\mathcal{M}_i(G_{\mathrm{pl}}))}{c - 1}$$
$$\alpha = (\beta - 1)P(\mathcal{M}_i(G_{\mathrm{pl}})) \qquad (\text{B.4})$$

*Proof.* Consider any set $S \subseteq \mathcal{M}_i(G_{\mathrm{pl}})$ with $P_i(S) > \frac{\beta P_i(\mathcal{M}_i(G_{\mathrm{pl}}))}{c-1}$. Then by Assumption 4.1, $P_i(\mathcal{N}(S)) \geq \min\{cP_i(S), 1\} \geq cP_i(S)$, where we used the fact that $P_i(S) \leq P_i(\mathcal{M}_i(G_{\mathrm{pl}}))$ and $c \leq \frac{1}{P_i(\mathcal{M}_i(G_{\mathrm{pl}}))}$, so $cP_i(S) \leq 1$. Thus, we can obtain

$$
\begin{aligned}
P_i(\mathcal{N}(S) \setminus \mathcal{M}_i(G_{\mathrm{pl}})) &\geq cP_i(S) - P_i(\mathcal{M}_i(G_{\mathrm{pl}})) \\
&= P_i(S) + (c-1)P_i(S) - P_i(\mathcal{M}_i(G_{\mathrm{pl}})) \\
&> P_i(S) + (\beta-1)P_i(\mathcal{M}_i(G_{\mathrm{pl}}))
\end{aligned}
$$

Here the last line used the fact that $P_i(S) > \frac{\beta P_i(\mathcal{M}_i(G_{\mathrm{pl}}))}{c-1}$. This implies that $P_i$ has $(q, \alpha)$-additive-expansion on $\mathcal{M}_i(G_{\mathrm{pl}})$ for the $(q, \alpha)$ given in (B.4). $\qquad\square$

We will now complete the proof of Lemma 4.2. Note that given Lemma 4.2, Theorem 4.3 follows immediately by noting that $G^\star$ satisfies $L_{0\text{-}1}(G^\star, G_{\mathrm{pl}}) = \mathrm{Err}(G_{\mathrm{pl}})$ and $R_{\mathcal{B}}(G^\star) \leq \mu$ by Assumption 3.3.

We first define the class-conditional pseudolabeling and robustness losses: $L_{0\text{-}1}^{(i)}(G, G') \triangleq P_i(\{x : G(x) \neq G'(x)\})$, and $R_{\mathcal{B}}^{(i)}(G) \triangleq \mathbb{E}_{P_i}[\mathbf{1}(\exists x' \in \mathcal{B}(x) \text{ such that } G(x') \neq G(x))]$. We also define the class-conditional error as follows: $\mathrm{Err}_i(G) \triangleq L_{0\text{-}1}^{(i)}(G, G^\star)$. We prove the class-conditional variant of Lemma 4.2 below.

**Lemma B.8.** *For any $i \in [K]$, define*

$$
\mathcal{L}_i(G) \triangleq \frac{c+1}{c-1}L_{0\text{-}1}^{(i)}(G, G_{\mathrm{pl}}) - \mathrm{Err}_i(G_{\mathrm{pl}}) + \frac{2c}{c-1}R_{\mathcal{B}}^{(i)}(G) \tag{B.5}
$$

*Then in the setting of Theorem 4.3 where Assumption 4.1 holds, we have the following error bound for class $i$:*

$$
\mathrm{Err}_i(G) \leq \mathcal{L}_i(G) \tag{B.6}
$$

*Proof.* First, we consider the case where $L_{0\text{-}1}^{(i)}(G, G_{\mathrm{pl}}) + R_{\mathcal{B}}^{(i)}(G) \leq (c-1)\mathrm{Err}_i(G_{\mathrm{pl}})$. In this case, we can apply Lemma B.7 with $\beta \in (0, c-1]$ chosen such that

$$
(\beta - 1)\mathrm{Err}_i(G_{\mathrm{pl}}) = L_{0\text{-}1}^{(i)}(G, G_{\mathrm{pl}}) + R_{\mathcal{B}}^{(i)}(G) - \mathrm{Err}_i(G_{\mathrm{pl}}) \tag{B.7}
$$

We note that $P_i$ has $(q, \alpha)$-additive-expansion on $\mathcal{M}_i(G_{\mathrm{pl}})$ for

$$
q = \frac{\beta}{c-1}\mathrm{Err}_i(G_{\mathrm{pl}}) \tag{B.8}
$$

$$
\alpha = (\beta - 1)\mathrm{Err}_i(G_{\mathrm{pl}}) \tag{B.9}
$$

Now by (B.7), we can apply Theorem B.2 with this choice of $(q, \alpha)$ to obtain

$$
\begin{aligned}
\mathrm{Err}_i(G) &\leq 2(q + R_{\mathcal{B}}^{(i)}(G)) + L_{0\text{-}1}^{(i)}(G, G_{\mathrm{pl}}) - \mathrm{Err}_i(G_{\mathrm{pl}}) & \text{(B.10)} \\
&= \frac{2\beta}{c-1}\mathrm{Err}_i(G_{\mathrm{pl}}) + 2R_{\mathcal{B}}^{(i)}(G) + L_{0\text{-}1}^{(i)}(G, G_{\mathrm{pl}}) - \mathrm{Err}_i(G_{\mathrm{pl}}) & \text{(B.11)} \\
&= \frac{c+1}{c-1}L_{0\text{-}1}^{(i)}(G, G_{\mathrm{pl}}) - \mathrm{Err}_i(G_{\mathrm{pl}}) + \frac{2c}{c-1}R_{\mathcal{B}}^{(i)}(G) & \text{(plugging in the value of } \beta) \\
&= \mathcal{L}_i(G) & \text{(B.12)}
\end{aligned}
$$

Next, we consider the case where $L_{0\text{-}1}^{(i)}(G, G_{\mathrm{pl}}) + R_{\mathcal{B}}^{(i)}(G) > (c-1)\mathrm{Err}_i(G_{\mathrm{pl}})$. Note that by triangle inequality, we have

$$
\begin{aligned}
\mathrm{Err}_i(G) = L_{0\text{-}1}^{(i)}(G, G^\star) &\leq L_{0\text{-}1}^{(i)}(G, G_{\mathrm{pl}}) + L_{0\text{-}1}^{(i)}(G_{\mathrm{pl}}, G^\star) & \text{(B.13)} \\
&= L_{0\text{-}1}^{(i)}(G, G_{\mathrm{pl}}) + 2\mathrm{Err}_i(G_{\mathrm{pl}}) - \mathrm{Err}_i(G_{\mathrm{pl}}) & \text{(B.14)} \\
&\leq L_{0\text{-}1}^{(i)}(G, G_{\mathrm{pl}}) + \frac{2}{c-1}(L_{0\text{-}1}^{(i)}(G, G_{\mathrm{pl}}) + R_{\mathcal{B}}^{(i)}(G)) - \mathrm{Err}_i(G_{\mathrm{pl}}) & \text{(B.15)} \\
&\leq \frac{c+1}{c-1}(L_{0\text{-}1}^{(i)}(G, G_{\mathrm{pl}}) + R_{\mathcal{B}}^{(i)}(G)) - \mathrm{Err}_i(G_{\mathrm{pl}}) & \text{(B.16)} \\
&\leq \frac{c+1}{c-1}L_{0\text{-}1}^{(i)}(G, G_{\mathrm{pl}}) + \frac{2c}{c-1}R_{\mathcal{B}}^{(i)}(G) - \mathrm{Err}_i(G_{\mathrm{pl}}) & \text{(using } c > 1) \\
&= \mathcal{L}_i(G) & \text{(B.17)}
\end{aligned}
$$

$\qquad\square$

Lemma 4.2 now follows simply by taking the expectation of the bound in (B.6) over all the classes.

## C  PROOFS FOR UNSUPERVISED LEARNING

We will first prove an analogue of Lemma C.7 for a relaxed notion of expansion. We will then prove Theorem 3.6 by showing that multiplicative expansion implies this relaxed notion, defined below:

**Definition C.1** ($(q, \xi)$-constant-expansion). *We say that distribution $P$ satisfies $(q, \xi)$-constant-expansion if for all $S \subseteq \mathcal{X}$ satisfying $P(S) \geq q$ and $P(S \cap \mathcal{C}_i) \leq P(\mathcal{C}_i)/2$ for all $i$, the following holds:*

$$P(\mathcal{N}^\star(S) \setminus S) \geq \min\{\xi, P(S)\}$$

As before, $\mathcal{N}^\star(S)$ is defined by $\cup_{i \in [K]} (\mathcal{N}(S \cap \mathcal{C}_i) \cap \mathcal{C}_i)$. We will work with the above notion of expansion for this subsection. We first show that a $\mathcal{B}$-robust labeling function which assigns sufficient probability to each class will align with the true classes.

**Theorem C.2.** *Suppose $P$ satisfies $(q, \xi)$-constant-expansion for some $q$. If it holds that $R_{\mathcal{B}}(G) < \xi$ and*

$$\min_i P(\{x : G(x) = i\}) > 2\max\{q, R_{\mathcal{B}}(G)\}$$

*there exists a permutation $\pi : [K] \to [K]$ satisfying the following:*

$$P(\{x : \pi(G(x)) \neq G^\star(x)\}) \leq \max\{q, R_{\mathcal{B}}(G)\} + R_{\mathcal{B}}(G) \tag{C.1}$$

Define $\widehat{\mathcal{C}}_1, \ldots, \widehat{\mathcal{C}}_K$ to be the partition induced by $G$: $\widehat{\mathcal{C}}_i \triangleq \{x : G(x) = i\}$.

**Lemma C.3.** *In the setting of Theorem C.2, consider any set of the form $U \triangleq \mathcal{S}_{\mathcal{B}}(G) \cap_{i \in \mathcal{I}} \mathcal{C}_i \cap_{j \in \mathcal{J}} \widehat{\mathcal{C}}_j$ where $\mathcal{I}, \mathcal{J}$ are arbitrary subsets of $[K]$. Then $\mathcal{N}^\star(U) \setminus U \subseteq \overline{\mathcal{S}_{\mathcal{B}}(G)}$.*

*Proof.* Consider any $x \in \mathcal{N}^\star(U) \setminus U$. There are two cases. First, if $G(x) \in \mathcal{J}$, then by definition of $\mathcal{N}^\star(\cdot)$, $x \in \cap_{i \in \mathcal{I}} \mathcal{C}_i \cap_{j \in \mathcal{J}} \widehat{\mathcal{C}}_j$. However, $x \notin U$, which must imply that $x \notin \mathcal{S}_{\mathcal{B}}(G)$. Second, if $G(x) \notin \mathcal{J}$, by definition of $\mathcal{N}^\star(\cdot)$ there exists $x' \in U$ such that $\mathcal{B}(x) \cap \mathcal{B}(x') \neq \emptyset$. It follows that for $z \in \mathcal{B}(x) \cap \mathcal{B}(x')$, $G(z) = G(x') \in \mathcal{J}$. Thus, since $G(x) \notin \mathcal{J}$, $G(x) \neq G(z)$ so $x \notin \mathcal{S}_{\mathcal{B}}(G)$. Thus, it follows that $\mathcal{N}^\star(U) \setminus U \subseteq \overline{\mathcal{S}_{\mathcal{B}}(G)}$. $\qquad\square$

Next, we show that every cluster found by $G$ will take up the majority of labels of some ground-truth class.

**Lemma C.4.** *In the setting of Theorem C.2, $\forall j$, $\exists i$ such that $P(\mathcal{S}_{\mathcal{B}}(G) \cap \mathcal{C}_i \cap \widehat{\mathcal{C}}_j) > \frac{P(\mathcal{S}_{\mathcal{B}}(G) \cap \mathcal{C}_i)}{2}$.*

*Proof.* Assume for the sake of contradiction that there exists $j$ such that for all $i$, $P(\mathcal{S}_{\mathcal{B}}(G) \cap \mathcal{C}_i \cap \widehat{\mathcal{C}}_j) \leq \frac{P(\mathcal{S}_{\mathcal{B}}(G) \cap \mathcal{C}_i)}{2}$. Define the set $U_i \triangleq \mathcal{S}_{\mathcal{B}}(G) \cap \mathcal{C}_i \cap \widehat{\mathcal{C}}_j$, and $U \triangleq \cup_i U_i = \mathcal{S}_{\mathcal{B}}(G) \cap \widehat{\mathcal{C}}_j$. Note that $\{U_i\}_{i=1}^K$ form a partition of $U$ because $\{\mathcal{C}_i\}_{i=1}^K$ are themselves disjoint from one another. Furthermore, we can apply Lemma C.3 with $\mathcal{I} = [K]$ to obtain $\mathcal{N}^\star(U) \setminus U \subseteq \overline{\mathcal{S}_{\mathcal{B}}(G)}$.

Now we observe that $P(U) \geq P(\widehat{\mathcal{C}}_j) - P(\overline{\mathcal{S}_{\mathcal{B}}(G)})$. Using the theorem condition that $P(\widehat{\mathcal{C}}_j) > 2P(\overline{\mathcal{S}_{\mathcal{B}}(G)})$, it follows that

$$P(U) > \frac{P(\widehat{\mathcal{C}}_j)}{2} > \max\{q, P(\overline{\mathcal{S}_{\mathcal{B}}(G)})\}$$

Furthermore for all $i$ we note that

$$P(\mathcal{C}_i \setminus U_i) \geq P(\mathcal{S}_{\mathcal{B}}(G) \cap \mathcal{C}_i) - P(U_i) \geq \frac{P(\mathcal{S}_{\mathcal{B}}(G) \cap \mathcal{C}_i)}{2} \geq P(U_i) \tag{C.2}$$

Thus, $P(\mathcal{C}_i) \geq 2P(U_i)$. Thus, by $(q, \xi)$-constant-expansion we have

$$P(\mathcal{N}^\star(U) \setminus U) \geq \min\{\xi, P(U)\} \geq \min\{\xi, P(\widehat{\mathcal{C}}_j)/2\}$$

As $\mathcal{N}^\star(U) \setminus U \subseteq \overline{\mathcal{S}_{\mathcal{B}}(G)}$, this implies $R_{\mathcal{B}}(G) = P(\overline{\mathcal{S}_{\mathcal{B}}(G)}) \geq \min\{\xi, P(\widehat{\mathcal{C}}_j)/2\}$, a contradiction. $\qquad\square$

**Lemma C.5.** *In the setting of Theorem C.2 and Lemma C.4, $\forall\, j$, there exists a unique $\pi(j)$ such that $P(\mathcal{S}_\mathcal{B}(G)\cap\mathcal{C}_{\pi(j)}\cap\widehat{\mathcal{C}}_j) > \frac{P(\mathcal{S}_\mathcal{B}(G)\cap\mathcal{C}_{\pi(j)})}{2}$, and $P(\mathcal{S}_\mathcal{B}(G)\cap\mathcal{C}_i\cap\widehat{\mathcal{C}}_j) \leq \frac{P(\mathcal{S}_\mathcal{B}(G)\cap\mathcal{C}_i)}{2}$ for $i\neq\pi(j)$. Furthermore, $\pi$ is a permutation from $[K]$ to $[K]$.*

*Proof.* By the conclusion of Lemma C.4, the only way the existence of such a $\pi$ might not hold is if there is some $j$ where $P(\mathcal{S}_\mathcal{B}(G)\cap\mathcal{C}_i\cap\widehat{\mathcal{C}}_j) > \frac{P(\mathcal{S}_\mathcal{B}(G)\cap\mathcal{C}_i)}{2}$ for $i\in\{i_1,i_2\}$, where $i_1\neq i_2$. In this case, by the Pigeonhole Principle, as the conclusion of Lemma C.4 applies for all $j\in[K]$ and there are $K$ possible choices for $i$, there must exist $i$ where $P(\mathcal{S}_\mathcal{B}(G)\cap\mathcal{C}_i\cap\widehat{\mathcal{C}}_j) > \frac{P(\mathcal{S}_\mathcal{B}(G)\cap\mathcal{C}_i)}{2}$ for $j\in\{j_1,j_2\}$, where $j_1\neq j_2$. Then $P(\mathcal{S}_\mathcal{B}(G)\cap\mathcal{C}_i\cap\widehat{\mathcal{C}}_{j_1}) + P(\mathcal{S}_\mathcal{B}(G)\cap\mathcal{C}_i\cap\widehat{\mathcal{C}}_{j_2}) > P(\mathcal{S}_\mathcal{B}(G)\cap\mathcal{C}_i)$, which is a contradiction.

Finally, to see that $\pi$ is a permutation, note that if $\pi(j_1) = \pi(j_2)$ for $j_1\neq j_2$, this would result in the same contradiction as above. $\qquad\square$

*Proof of Theorem C.2.* We will prove (C.1) using $\pi$ defined in Lemma C.5. Define the set $U_j \triangleq \mathcal{S}_\mathcal{B}(G)\cap\mathcal{C}_{\pi(j)}\cap_{k\neq j}\widehat{\mathcal{C}}_k$. Note that $U_j = \{x : G(x)\neq j, G^\star(x) = \pi(j)\}\cap\mathcal{S}_\mathcal{B}(G)$. Define $U = \cup_j U_j$, and note that $\{U_j\}_{j=1}^K$ forms a partition of $U$. Furthermore, we also have $U = \{x : \pi(G(x))\neq G^\star(x)\}\cap\mathcal{S}_\mathcal{B}(G)$. We first show that $P(U)\leq\max\{q, R_\mathcal{B}(G)\}$. Assume for the sake of contradiction that this does not hold.

First, we claim that $\{\mathcal{N}^\star(U_j)\setminus U_j\}_{j=1}^k \supseteq \mathcal{N}^\star(U)\setminus U$. To see this, consider any $x\in\mathcal{C}_{\pi(j)}\cap\mathcal{N}^\star(U)\setminus U$. By definition, $\exists x'\in U$ such that $\mathcal{B}(x')\cap\mathcal{B}(x)\neq\emptyset$ and $G^\star(x) = G^\star(x')$, or $x'\in\mathcal{C}_{\pi(j)}$. Thus, it follows that $x\in\mathcal{N}^\star(\mathcal{C}_{\pi(j)}\cap U)\setminus U = \mathcal{N}^\star(U_j)\setminus U = \mathcal{N}^\star(U_j)\setminus U_j$, where the last equality followed from the fact that $\mathcal{N}^\star(U_j)$ and $U_k$ are disjoint for $j\neq k$. Now we apply Lemma C.3 to each $\mathcal{N}^\star(U_j)\setminus U_j$ to conclude that $\mathcal{N}^\star(U)\setminus U\subseteq\overline{\mathcal{S}_\mathcal{B}(G)}$.

Finally, we observe that

$$P(U_j) = P(\mathcal{S}_\mathcal{B}(G)\cap\mathcal{C}_{\pi(j)}) - P(\mathcal{S}_\mathcal{B}(G)\cap\mathcal{C}_{\pi(j)}\cap\widehat{\mathcal{C}}_j) \leq \frac{P(\mathcal{S}_\mathcal{B}(G)\cap\mathcal{C}_{\pi(j)})}{2} \leq \frac{P(\mathcal{C}_{\pi(j)})}{2} \quad\text{(C.3)}$$

by the definition of $\pi$ in Lemma C.5. Now we again apply the $(q,\xi)$-constant-expansion property, as we assumed $P(U) > q$, obtaining

$$P(\mathcal{N}^\star(U)\setminus U) \geq \min\{\xi, P(U)\}$$

However, as we showed $\mathcal{N}^\star(U)\setminus U\subseteq\overline{\mathcal{S}_\mathcal{B}(G)}$, we also have $R_\mathcal{B}(G) = P(\overline{\mathcal{S}_\mathcal{B}(G)}) \geq P(\mathcal{N}^\star(U)\setminus U) \geq \min\{\xi, P(U)\}$. This contradicts $P(U) > \max\{q, R_\mathcal{B}(G)\}$ and $R_\mathcal{B}(G) < \xi$, and therefore $P(U)\leq\max\{q, R_\mathcal{B}(G)\}$.

Finally, we note that $\{x : \pi(G(x))\neq G^\star(x)\}\subseteq U\cup\overline{\mathcal{S}_\mathcal{B}(G)}$. Thus, we finally obtain

$$P(\{x : \pi(G(x))\neq G^\star(x)\}) \leq P(U) + P(\overline{\mathcal{S}_\mathcal{B}(G)}) \leq \max\{q, R_\mathcal{B}(G)\} + R_\mathcal{B}(G)$$

$\qquad\square$

## C.1    Proof of Theorem 3.6

In this section, we prove Theorem 3.6 by converting multiplicative expansion to $(q,\xi)$-constant-expansion and invoking Theorem C.2. The following lemma performs this conversion.

**Lemma C.6.** *Suppose $P$ satisfies $(1/2, c)$-multiplicative-expansion (Definition 3.1) on $\mathcal{X}$. Then for any choice of $\xi > 0$, $P$ satisfies $\left(\frac{\xi}{c-1}, \xi\right)$-constant expansion.*

*Proof.* Consider any $S$ such that $P(S \cap C_i) \leq P(C_i)/2$ for all $i \in [K]$ and $P(S) > q$. Define $S_i \triangleq S \cap C_i$. First, in the case where $c \geq 2$, we have by multiplicative expansion

$$
\begin{aligned}
P(\mathcal{N}^\star(S) \setminus S) &\geq \sum_i P(\mathcal{N}^\star(S_i)) - P(S_i) \\
&\geq \sum_i \min\{cP(S_i), P(C_i)\} - P(S_i) \\
&\geq \sum_i P(S_i) \qquad \text{(because } c \geq 2 \text{ and } P(S_i) \leq P(C_i)/2\text{)}
\end{aligned}
$$

Thus, we immediately obtain constant expansion.

Now we consider the case where $1 \leq c < 2$. By multiplicative expansion, we must have

$$
\begin{aligned}
P(\mathcal{N}^\star(S) \setminus S) &\geq \sum_i \min\{cP(S_i), P(C_i)\} - P(S_i) \\
&\geq \sum_i (c-1)P(S_i) \qquad \text{(because } c < 2 \text{ and } P(S_i) \leq P(C_i)/2\text{)} \\
&\geq (c-1)q = \xi
\end{aligned}
$$

$\square$

The following lemma states an accuracy guarantee for the setting with multiplicative expansion.

**Lemma C.7.** *Suppose Assumption 3.2 holds for some $c > 1$. If classifier $G$ satisfies*

$$
\min_i \mathbb{E}_P[\mathbf{1}(G(x) = i)] > \max\left\{\frac{2}{c-1}, 2\right\} R_{\mathcal{B}}(G)
$$

*then the unsupervised error is small:*

$$
\text{Err}_{\text{unsup}}(G) \leq \max\left\{\frac{c}{c-1}, 2\right\} R_{\mathcal{B}}(G) \tag{C.4}
$$

We now prove Lemma C.7, which in turn immediately gives a proof of Theorem 3.6.

*Proof of Lemma C.7.* By Lemma C.6, $P$ must satisfy $\left(\frac{R_{\mathcal{B}}(G)}{c-1}, R_{\mathcal{B}}(G)\right)$-constant-expansion. As we also have $\min_i P(\{x : G(x) = i\}) > \max\left\{\frac{2}{c-1}, 2\right\} R_{\mathcal{B}}(G)$, we can now apply Theorem C.2 to conclude that there exists permutation $\pi : [K] \to [K]$ such that

$$
P(\{x : \pi(G(x)) \neq G^\star(x)\}) \leq \max\left\{\frac{c}{c-1}, 2\right\} R_{\mathcal{B}}(G)
$$

as desired. $\square$

## C.2  JUSTIFICATION FOR EXAMPLES 3.4 AND 3.5

To avoid the disjointness issue of Example 3.4, we can redefine the ground-truth class $G^\star(x)$ to be the most likely label at $x$. This also induces truncated class-conditional distributions $\overline{P}_1, \overline{P}_2$ where the overlap is removed. We can apply our theoretical analysis to $\overline{P}_1, \overline{P}_2$ and then translate the result back to $P_1, P_2$, only changing the bounds by a small amount when the overlap is minimal.

To justify Example 3.4, we use the Gaussian isoperimetric inequality (Bobkov et al., 1997), which states that for any fixed $p$ such that $P_i(S) = p$ where $i \in \{1, 2\}$, the choice of $S$ minimizing $P_i(\mathcal{N}(S))$ is given by a halfspace: $S = H(p) \triangleq \{x : w^\top(x - \tau_i) \leq \Phi^{-1}(p)\}$ for vector $w$ with $\|w\| = \sqrt{d}$. It then follows that setting $r = \frac{1}{\sqrt{d}}$, $\mathcal{N}(H(p)) \supseteq \{x + t\frac{w}{\|w\|_2} : x \in H(p), 0 \leq t \leq r\} \supseteq \{x : w^\top(x - \tau_i) \leq \Phi^{-1}(p) + r\sqrt{d}\}$, and thus $P(\mathcal{N}(H(p))) \geq \Phi(\Phi^{-1}(p) + r\sqrt{d})$. As $P(\mathcal{N}(H(p)))/P(H(p))$ is decreasing in $p$ for $p < 0.5$, our claim about expansion follows. To see

our claim about separation, consider the sets $\mathcal{X}_i \triangleq \{x : (x - \tau_i)^\top v_{ij} \leq \frac{\|\tau_i - \tau_j\|}{2} - r/2 \ \forall j\}$, where $v_{ij} \triangleq \frac{\tau_j - \tau_i}{\|\tau_j - \tau_i\|_2}$. We note that these sets are $\beta$-separated from each other, and furthermore, for the lower bound on $\|\tau_i - \tau_j\|$ in the example, note that $\mathcal{X}_i$ has probability $1 - \mu$ under $P_i$.

For Example 3.5, we note that for $\mathcal{B}(x) \triangleq \{x' : \|x' - x\|_2 \leq r\}$, $\mathcal{N}(S) \supseteq M(\{x' : \exists x \in M^{-1}(S) \text{ such that } \|x' - x\| \leq r/\kappa\})$. Thus, our claim about expansion reduces to the Gaussian case.

# D  ALL-LAYER MARGIN GENERALIZATION BOUNDS

## D.1  END-TO-END GUARANTEES

In this section, we provide end-to-end guarantees for unsupervised learning, semi-supervised learning, and unsupervised domain adaptation for finite training sets. For the following two theorems, we take the notation $\tilde{O}(\cdot)$ as a placeholder for some multiplicative quantity that is poly-logarithmic in $n, d$. We first provide the finite-sample guarantee for unsupervised learning.

**Theorem D.1.** *In the setting of Theorem 3.6 and Section 3.3, suppose that Assumption 3.2 holds. Suppose that $G = \arg\max_i F_i$ is parametrized as a neural network of the form $F(x) \triangleq W_p \phi(\cdots \phi(W_1 x) \cdots)$. With probability $1 - \delta$ over the draw of the training sample $\widehat{P}$, if for any choice of $t > 0$ and $\{u_y\}_{y=1}^K$ with $u_y > 0 \ \forall y$, it holds that*

$$\mathbb{E}_{\widehat{P}}[\mathbf{1}(m(F, x, y) \geq u_y)] - \max\left\{\frac{2}{c-1}, 2\right\} \mathbb{E}_{\widehat{P}}[\mathbf{1}(m_\mathcal{B}(F, x) \leq t)]$$

$$\geq \tilde{O}\left(\left(\frac{\sum_i \sqrt{q}\|W_i\|_F}{c-1}\right)\left(\frac{1}{u_y \sqrt{n}} + \frac{1}{t \sqrt{n}}\right)\right) + \zeta \quad \text{for all } y \in [K]$$

*then it follows that the population unsupervised error is small:*

$$\text{Err}_{\text{unsup}}(G) \leq \max\left\{\frac{c}{c-1}, 2\right\} \mathbb{E}_{\widehat{P}}[\mathbf{1}(m_\mathcal{B}(F, x) \leq t)] + \tilde{O}\left(\frac{\sum_i \sqrt{q}\|W_i\|_F}{t \sqrt{n}}\right) + \zeta$$

*where $\zeta \triangleq O\left(\frac{1}{c-1}\sqrt{\frac{\log(K/\delta) + p \log n}{n}}\right)$ is a low-order term.*

The following theorem provides the finite-sample guarantee for unsupervised domain adaptation and semi-supervised learning.

**Theorem D.2.** *In the setting of Theorem 4.3 and Section 3.3, suppose that Assumption 4.1 holds. Suppose that $G = \arg\max_i F_i$ is parametrized as a neural network of the form $F(x) \triangleq W_p \phi(\cdots \phi(W_1 x) \cdots)$. For any $t_1, t_2 > 0$, define the following quantities:*

$$B_1 \triangleq 2\mathbb{E}_{\widehat{P}}[\mathbf{1}(m_\mathcal{B}(F, x) \leq t_1)] + \mathbb{E}_{\widehat{P}}[\mathbf{1}(m(F, x, G_{\text{pl}}(x)) \leq t_2)]$$

$$+ \tilde{O}\left(\left(\sum_i \sqrt{q}\|W_i\|_F\right)\left(\frac{1}{t_1 \sqrt{n}} + \frac{1}{t_2 \sqrt{n}}\right)\right) + \zeta$$

$$B_2 \triangleq 4\mathbb{E}_{\widehat{P}}[\mathbf{1}(m_\mathcal{B}(F, x) \leq t_1)] + 3\mathbb{E}_{\widehat{P}}[\mathbf{1}(m(F, x, G_{\text{pl}}(x)) \leq t_2)]$$

$$+ \tilde{O}\left(\left(\sum_i \sqrt{q}\|W_i\|_F\right)\left(\frac{1}{t_1 \sqrt{n}} + \frac{1}{t_2 \sqrt{n}}\right)\right) + \zeta$$

*where $\zeta \triangleq O\left(\frac{1}{c-1}\sqrt{\frac{\log(K/\delta) + p \log n}{n}}\right)$ is a low-order term. With probability $1 - \delta$ over the draw of the training sample $\widehat{P}$, for all choices of $t_1, t_2 > 0$, it holds that*

$$\text{Err}(G) \leq \max\left\{B_1 - \text{Err}(G_{\text{pl}}), B_2 - \left(3 - \frac{4}{c-1}\right)\text{Err}(G_{\text{pl}})\right\}$$

### D.2 PROOFS FOR SECTION 3.3

In this section, we provide a proof sketch of Theorem 3.7. The proof follows the analysis of (Wei & Ma, 2019b) very closely, but because there are some minor differences we include it here for completeness. We first state additional bounds for the other quantities in our objectives, which are proved in the same manner as Theorem 3.7.

**Theorem D.3.** *With probability $1 - \delta$ over the draw of the training sample $\widehat{P}$, all neural networks $G = \arg\max_i F_i$ of the form $F(x) \triangleq W_p\phi(\cdots\phi(W_1x))$ will satisfy*

$$L_{\text{0-1}}(G, G_{\text{pl}}) \leq \mathbb{E}_{\widehat{P}}[\mathbf{1}(m(F, x, G_{\text{pl}}(x)) \leq t)] + \widetilde{O}\left(\frac{\sum_i \sqrt{q}\|W_i\|_F}{t\sqrt{n}}\right) + \zeta$$

*for all choices of $t > 0$, where $\zeta \triangleq O\left(\sqrt{\frac{\log(1/\delta) + p\log n}{n}}\right)$ is a low-order term, and $\widetilde{O}(\cdot)$ hides poly-logarithmic factors in $n$ and $d$.*

**Theorem D.4.** *With probability $1 - \delta$ over the draw of the training sample $\widehat{P}$, all neural networks $G = \arg\max_i F_i$ of the form $F(x) \triangleq W_p\phi(\cdots\phi(W_1x))$ will satisfy*

$$\mathbb{E}_P[\mathbf{1}(G(x) = y)] \geq \mathbb{E}_{\widehat{P}}[\mathbf{1}(m(F, x, y) \geq t)] - \widetilde{O}\left(\frac{\sum_i \sqrt{q}\|W_i\|_F}{t\sqrt{n}}\right) - \zeta$$

*for all choices of $y \in [K]$, $t > 0$, where $\zeta \triangleq O\left(\sqrt{\frac{\log(K/\delta) + p\log n}{n}}\right)$ is a low-order term, and $\widetilde{O}(\cdot)$ hides poly-logarithmic factors in $n$ and $d$.*

We now overview the proof of Theorem 3.7, as the proofs of Theorem D.3 and D.4 follow identically. We first formally define the all-layer margin $m(F, x, y)$ for neural net $F$ evaluated on example $x$ with label $y$. We recall that $F$ computes the function $F(x) \triangleq W_p\phi(\cdots\phi(W_1x)\cdots)$. We index the layers of $F$ as follows: define $f_1(x) \triangleq W_1x$, and $f_i(h) \triangleq W_i\phi(h)$ for $2 \leq i \leq p$, so that $F(x) = f_p \circ \cdots \circ f_1(x)$. Letting $\delta = (\delta_1, \ldots, \delta_p)$ denote perturbations for each layer of $F$, we define the perturbed output $F(x, \delta)$ as follows:

$$h_1(x, \delta) = f_1(x) + \delta_1\|x\|_2$$
$$h_i(x, \delta) = f_i(h_{i-1}(x, \delta)) + \delta_i\|h_{i-1}(x, \delta)\|_2$$
$$F(x, \delta) = h_p(x, \delta)$$

Now the all-layer margin $m(F, x, y)$ is defined by

$$m(F, x, y) \triangleq \quad \min_{\delta} \sqrt{\sum_{i=1}^{p} \|\delta_i\|_2^2}$$
$$\text{subject to } \arg\max_i F(x, \delta) \neq y$$

As is typical in generalization bound proofs, we define a fixed class of neural net functions to analyze, expressed as

$$\mathcal{F} \triangleq \{x \mapsto W_p\phi(\cdots\phi(W_1x)\cdots) : W_i \in \mathcal{W}_i \ \forall i\}$$

where $\mathcal{W}_i$ is some class of possible instantiations of the $i$-th weight matrix. We also overload notation and let $\mathcal{W}_i \triangleq \{h \mapsto W_ih : W_i \in \mathcal{W}_i\}$ denote the class of functions corresponding to matrix multiplication by a weight in $\mathcal{W}_i$. Let $\|\cdot\|_{\text{op}}$ denote the matrix operator norm. For a function class $\mathcal{G}$, we let $\mathcal{N}_{\|\cdot\|}(\epsilon, \mathcal{G})$ denote the $\epsilon$-covering number of $\mathcal{G}$ in norm $\|\cdot\|$. The following condition will be useful for the analysis:

**Condition D.5** (Condition A.1 from (Wei & Ma, 2019b)). *We say that a function class $\mathcal{G}$ satisfies the $\epsilon^{-2}$ covering condition with respect to norm $\|\cdot\|$ with complexity $\mathcal{C}_{\|\cdot\|}(\mathcal{G})$ if for all $\epsilon > 0$,*

$$\log\mathcal{N}_{\|\cdot\|}(\epsilon, \mathcal{G}) \leq \left\lfloor \frac{\mathcal{C}_{\|\cdot\|}^2(\mathcal{G})}{\epsilon^2} \right\rfloor$$

To sketch the proof technique, we only provide the proof of (3.6) in Theorem 3.7, as the other bounds follow with the same argument. The following lemma bounds $R_{\mathcal{B}}(G)$ in terms of the robust all-layer margin $m_{\mathcal{B}}$.

**Lemma D.6** (Adaptation of Theorem A.1 of (Wei & Ma, 2019b))**.** *Suppose that weight matrix mappings $\mathcal{W}_i$ satisfy Condition D.5 with operator norm $\|\cdot\|_{\mathrm{op}}$ and complexity function $\mathcal{C}_{\|\cdot\|_{\mathrm{op}}}(\mathcal{W}_i)$. With probability $1 - \delta$ over the draw of the training data, for all $t > 0$, all classifiers $F \in \mathcal{F}$ will satisfy*

$$R_{\mathcal{B}}(G) \leq \mathbb{E}_{\widehat{P}}[\mathbf{1}(m_{\mathcal{B}}(F, x) \leq t)] + O\left(\frac{\sum_i \mathcal{C}_{\|\cdot\|_{\mathrm{op}}}(\mathcal{W}_i)}{t\sqrt{n}} \log n\right) + \zeta \tag{D.1}$$

*where $\zeta \triangleq O\left(\sqrt{\frac{\log(1/\delta) + \log n}{n}}\right)$ is a low-order term.*

The proof of Lemma D.6 mirrors the proof of Theorem A.1 of (Wei & Ma, 2019b). The primary difference is that because we seek a bound in terms a threshold on the margin whereas (Wei & Ma, 2019b) prove a bound that depends on average margin, we must analyze the generalization of a slightly modified loss. Towards proving Lemma D.6, we first define $\|\!|\delta|\!\| \triangleq \|(\|\delta_1\|_2, \ldots, \|\delta_p\|_2)\|_2$ for perturbation $\delta$, and $\|\!|F|\!\| \triangleq \|(\|W_1\|_{\mathrm{op}}, \ldots, \|W_p\|_{\mathrm{op}})\|_2$. We show that $m_{\mathcal{B}}(F, x)$ is Lipschitz in $F$ for fixed $x$ with respect to $\|\!|\cdot|\!\|$.

**Claim D.7.** *Choose $F, \widehat{F} \in \mathcal{F}$. Then for any $x \in \mathcal{X}$,*

$$|m_{\mathcal{B}}(F, x) - m_{\mathcal{B}}(\widehat{F}, x)| \leq \|\!|F - \widehat{F}|\!\|$$

*The same conclusion holds if we replace $m_{\mathcal{B}}$ with $m$.*

*Proof.* We consider two cases:

Case 1: $\arg\max_i F(x)_i = \arg\max_i \widehat{F}(x)_i$. Let $y$ denote the common value. In this case, the desired result immediately follows from Claim E.1 of (Wei & Ma, 2019b).

Case 2: $\arg\max_i F(x)_i \neq \arg\max_i \widehat{F}(x)_i$. In this case, the construction of Claim A.1 in (Wei & Ma, 2019b) implies that $0 \leq m_{\mathcal{B}}(F, x) \leq \|\!|F - \widehat{F}|\!\|$. (Essentially we choose $\delta$ with $\|\!|\delta|\!\| \leq \|\!|F - \widehat{F}|\!\|$ such that $F(x, \delta) = \widehat{F}(x)$.) Likewise, $0 \leq m_{\mathcal{B}}(\widehat{F}, x) \leq \|\!|F - \widehat{F}|\!\|$. As a result, it must follow that $|m_{\mathcal{B}}(F, x) - m_{\mathcal{B}}(\widehat{F}, x)| \leq \|\!|F - \widehat{F}|\!\|$. $\square$

For $t > 0$, define the ramp loss $h_t$ as follows:

$$h_t(a) = 1 - \mathbf{1}(a \geq 0)\min\{a/t, 1\}$$

We now define the hypothesis class $\mathcal{L}_t \triangleq \{h_t \circ m_{\mathcal{B}}(F, \cdot) : F \in \mathcal{F}\}$. We now bound the Rademacher complexity of this hypothesis class:

**Claim D.8.** *In the setting of Lemma D.6, suppose that $\mathcal{W}_i$ satisfies Condition D.5 with operator norm $\|\cdot\|_{\mathrm{op}}$ and complexity $\mathcal{C}_{\|\cdot\|_{\mathrm{op}}}(\mathcal{W}_i)$. Then*

$$\mathrm{Rad}_n(\mathcal{L}_t) \leq O\left(\frac{\sum_i \mathcal{C}_{\|\cdot\|_{\mathrm{op}}}(\mathcal{W}_i)}{t\sqrt{n}} \log n\right)$$

As the proof of Claim D.8 is standard, we provide a sketch of its proof.

*Proof sketch of Claim D.8.* First, by Lemma A.3 of (Wei & Ma, 2019b), we obtain that $\mathcal{F}$ satisfies Condition D.5 with norm $\|\!|\cdot|\!\|$ and complexity $\mathcal{C}_{\|\!|\cdot|\!\|}(\mathcal{F}) \triangleq \sum_i \mathcal{C}_{\|\cdot\|_{\mathrm{op}}}(\mathcal{F}_i)$. Now let $\widehat{\mathcal{F}}$ be a $t\epsilon$-cover of $\mathcal{F}$ in $\|\!|\cdot|\!\|$. We define the $L_2(P_n)$-norm of a function $f : \mathcal{X} \to \mathbb{R}$ as follows:

$$\|f\|_{L_2(P_n)} \triangleq \sqrt{\mathbb{E}_{\widehat{P}}[f(x)^2]}$$

Then it is standard to show that

$$\widehat{\mathcal{L}}_t \triangleq \{h_t \circ m_{\mathcal{B}}(\widehat{F}, \cdot) : \widehat{F} \in \widehat{\mathcal{F}}\}$$

is a $\epsilon$-cover of $\mathcal{L}_t$ in $L_2(P_n)$-norm, because $h_t$ is $1/t$-Lipschitz and $m_{\mathcal{B}}(F, x)$ is 1-Lipschitz in $F$ for norm $\|\!|\cdot\|\!|$ for any fixed $x$. It follows that $\log \mathcal{N}_{L_2(P_n)}(\epsilon, \mathcal{L}_t) \leq \left\lfloor \frac{\mathcal{C}^2_{\|\!|\cdot\|\!|}(\mathcal{F})}{t^2 \epsilon^2} \right\rfloor$. Now we apply Dudley's Theorem:

$$\text{Rad}_n(\mathcal{L}_t) \leq \inf_{\beta > 0} \left( \beta + \frac{1}{\sqrt{n}} \int_\beta^\infty \sqrt{\log \mathcal{N}_{L_2(P_n)}(\epsilon, \mathcal{L}_t)} d\epsilon \right)$$

$$\leq \inf_{\beta > 0} \left( \beta + \frac{1}{\sqrt{n}} \int_\beta^\infty \sqrt{\left\lfloor \frac{\mathcal{C}^2_{\|\!|\cdot\|\!|}(\mathcal{F})}{t^2 \epsilon^2} \right\rfloor} d\epsilon \right)$$

A standard computation can be used to bound the quantity on the right, giving the desired result. □

*Proof of Lemma D.6.* First, by the standard relationship between Rademacher complexity and generalization, Claim D.8 lets us conclude that with probability $1 - \delta$, for any *fixed* $t > 0$, all $F \in \mathcal{F}$ satisfy:

$$\mathbb{E}_P[h_t(m_{\mathcal{B}}(F, x))] \leq \mathbb{E}_{\widehat{P}}[h_t(m_{\mathcal{B}}(F, x))] + O\left( \frac{\sum_i \mathcal{C}_{\|\cdot\|_{\text{op}}}(\mathcal{W}_i)}{t\sqrt{n}} \log n + \sqrt{\frac{\log 1/\delta}{n}} \right)$$

We additionally note that $h_t(m_{\mathcal{B}}(F, x)) = 1$ when $x \notin \mathcal{S}_{\mathcal{B}}(G)$, because in such cases $m_{\mathcal{B}}(F, x) = 0$. It follows that $1(x \notin \mathcal{S}_{\mathcal{B}}(G)) \leq h_t(m_{\mathcal{B}}(F, x))$. Thus, we obtain

$$R_{\mathcal{B}}(G) \leq \mathbb{E}_{\widehat{P}}[\mathbf{1}(m_{\mathcal{B}}(F, x) \leq t)] + O\left( \frac{\sum_i \mathcal{C}_{\|\cdot\|_{\text{op}}}(\mathcal{W}_i)}{t\sqrt{n}} \log n + \sqrt{\frac{\log 1/\delta}{n}} \right) \qquad \text{(D.2)}$$

It remains to show that (D.1) holds for *all* $t$. It is now standard to perform a union bound over choices of $t$ in the form $t_j \triangleq t_{\min} 2^j$, where $t_{\min} \triangleq \frac{\sum_i \mathcal{C}_{\|\cdot\|_{\text{op}}}(\mathcal{W}_i)}{\sqrt{n}} \log n$ and $0 \leq j \leq O(\log n)$, so we only sketch the argument here. We union bound over (D.2) for $t = t_j$ with failure probability $\delta_j = \delta/2^{j+1}$, so (D.2) will hold for all $t_1, \ldots, t_{j_{\max}}$ with probability $1 - \delta$. For any choice of $t$, there will either be $j$ such that $t/2 \leq t_j \leq t$, or (D.1) must trivially hold. (See Theorem C.1 of (Wei & Ma, 2019b) for a more detailed justification.) As a result, there will be some $j$ such that the right hand side of (D.2) is bounded above by the right hand side of (D.1), as desired. □

*Proof sketch of Theorem 3.7.* By Lemma B.2 of (Wei & Ma, 2019b), we have $\mathcal{C}_{\|\cdot\|_{\text{op}}}(\{W : \|W\|_F \leq a\}) = O(\sqrt{q \log q} a)$. Thus, to obtain (3.6), it suffices to apply Lemma D.6 for all choices of $a$ using a standard union bound technique; see for example the proof of Theorem 3.1 in (Wei & Ma, 2019b). To obtain the other generalization bounds, we can follow a similar argument for Lemma D.6 to prove its analogue for other variants of all-layer margin, and then repeat the same union bound over the weight matrix norms as before. □

### D.3 DATA-DEPENDENT LOWER BOUNDS ON ALL-LAYER MARGIN

We will now provide lower bounds on the all-layer margins used in Theorem 3.7 in the case when the activation $\phi$ has $\overline{\nu}$-Lipschitz derivative. In this section, it will be convenient to modify the indexing to count the activation as its own layer, so there are $2p - 1$ layers in total. Let $s_{(i)}(x)$ denote the $\|\cdot\|_2$ norm of the layer preceding the $i$-th matrix multiplication, where the parenthesis in the subscript distinguishes between weight indices and layer indices (which also include the activation layers). Define $\nu_{j \leftarrow i}(x)$ to be the Jacobian of the $j$-th layer with respect to the $i - 1$-th layer evaluated at $x$. Define $\gamma(F(x), y) \triangleq F(x)_y - \max_{i \neq y} F(x)_i$. We use the following quantity to measure stability in the layer following $W_{(i)}$:

$$\kappa_{(i)}(x, y) \triangleq \frac{s_{(i-1)}(x) \nu_{2p-1 \leftarrow 2i}(x)}{\gamma(F(x), y)} + \psi_{(i)}(x, y)$$

for a secondary term $\psi_{(i)}(x, y)$ given by

$$\psi_{(i)}(x, y) \triangleq \sum_{j=i}^{p-1} \frac{s_{(i-1)}(x)\nu_{2j \leftarrow 2i}(x)}{s_{(j)}(x)} + \sum_{1 \le j \le 2i-1 \le j' \le 2p-1} \frac{\nu_{j' \leftarrow 2i}(x)\nu_{2i-2 \leftarrow j}(x)}{\nu_{j' \leftarrow j}(x)}$$

$$+ \sum_{1 \le j \le j' \le 2p-1} \sum_{j'' = \max\{2i,j\}, j'' \text{even}}^{j'} \frac{\overline{\nu}\nu_{j' \leftarrow j''+1}(x)\nu_{j''-1 \leftarrow 2i}(x)\nu_{j''-1 \leftarrow j}(x)s_{(i-1)}(x)}{\nu_{j' \leftarrow j}(x)}$$

We now have the following lower bounds on $m(F, x, y)$ and $m_{\mathcal{B}}(F, x)$:

**Proposition D.9** (Lemma C.1 from (Wei & Ma, 2019b))**.** *In the setting above, if $\gamma(F(x), y) > 0$, we have*

$$m(F, x, y) \ge \frac{1}{\|\{\kappa_{(i)}(x, y)\}_{i=1}^{p}\|_2}$$

*Furthermore, if $\gamma(F(x'), \arg\max_i F(x)_i) > 0$ for all $x' \in \mathcal{B}(x)$, then*

$$m_{\mathcal{B}}(F, x) \ge \min_{x' \in \mathcal{B}(x)} \frac{1}{\|\{\kappa_{(i)}(x', \arg\max_i F(x)_i)\}_{i=1}^{p}\|_2}$$

# E  EXPERIMENTS

## E.1  EMPIRICAL SUPPORT FOR EXPANSION PROPERTY USING GANS

In this section we provide additional details regarding the GAN verification depicted in Figure 1 (left). We use 128 by 128 images sampled from a pre-trained BigGAN (Brock et al., 2018). We categorize images into 10 superclasses chosen in the robustness library of Engstrom et al. (2019): dog, bird, insect, monkey, car, cat, truck, fruit, fungus, boat. These superclasses consist of all ImageNet classes which fall under the category of the superclass. To sample an image from a superclass, we uniformly sample an ImageNet class from the superclass and then sample from the GAN conditioned on this class. We sample 1000 images per superclass and train a ResNet-56 (He et al., 2016) to predict the superclass, achieving 93.74% validation accuracy.

Next, we approximately project GAN images onto the mislabeled set of the trained classifier. We approximate the projection as follows: we optimize an objective consisting of the $\ell_2$ distance from the original image and the negative cross entropy loss of the pretrained classifier w.r.t the superclass label. Letting $M$ denote the GAN mapping, $x$ the original image, $y$ the label, and $F$ the pre-trained classifier, the objective is as follows:

$$\min_z \|x - M(z)\|_2^2 - \lambda_{\text{ce}} \ell_{\text{cross-ent}}(F(M(z)), y)$$

We optimize $z$ for 2000 gradient descent steps using $\lambda_{\text{ce}} = 10$ and a learning rate of 0.0003, intialized with the same latent variable as was used to generate $x$. The resulting $M(z)$ is a neighbor of $x$ in the set $\mathcal{M}(F)$, the mistakenly labeled set of $F$.

After performing this procedure on 200 GAN images sampled from each class, we find that 20% of these images $x$ have a neighbor $x' \in \mathcal{M}(F)$ with $\|x - x'\|_2 \le 19.765$. Note that this corresponds to modifying each pixel by 0.024 on average for pixel values in [0, 1]. We use $\widehat{\mathcal{M}}$ to denote the set of mislabeled neighbors found this way. From visual inspection, we find that the neighbors appear very visually similar to the original image, suggesting that it is appropriate to regard these images as "neighbors". In Figure 1, we visualize typical examples of the neighbors found by this procedure. Thus, setting $\mathcal{B}(x) = \{x' : \|x' - x\|_2 \le \frac{19.765}{2}\}$, the set $\mathcal{M}(F)$, which has probability 0.0626, has a relatively large neighborhood induced by $\mathcal{B}$ of probability 0.2. This supports our expansion assumption, especially the additive notion in Section B.

Next, we use this same classifier as a pseudolabeler to perform self-training on a dataset of 10000 additional unlabeled images per superclass, where these images were sampled independently from the 200 GAN images in the previous step. We add input consistency regularization to the self-training procedure using VAT (Miyato et al., 2018). After self-training, the validation accuracy of new classifier $\widetilde{G}$ improves to 95.69%.

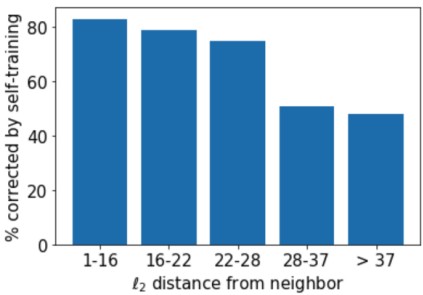

Figure 3: **Self-training corrects mistakenly labeled examples that are close to correctly labeled neighbors.** We partition examples in $\widehat{\mathcal{M}}'$ (defined in Section E.1) into 5 bins based on their $\ell_2$ distance from the neighbor used to initialize the projection, and plot the percentage of examples in each bin whose labels were corrected by self-training. The bins are chosen to be equally sized. The plot suggests that as a mistakenly labeled example is closer to a correctly labeled example in input space, it is more likely to be corrected by self-training. This supports our theoretical intuition that input-consistency-regularized self-training denoises pseudolabels by bootstrapping an incorrectly pseudolabeled example with its correctly pseudolabeled neighbors.

Furthermore, we evaluate performance of the self-trained classifier $\widetilde{G}$ on a subset of $\widehat{\mathcal{M}}$ with distance greater than 1 from its neighbor. We let $\widehat{\mathcal{M}}'$ denote this subset. We choose to filter $\widehat{\mathcal{M}}$ this way to rule out cases where the original neighbor was already misclassified. We find that $\widetilde{G}$ achieves 67.27% accuracy on examples from $\widehat{\mathcal{M}}'$.

In addition, Figure 3 demonstrates that $\widetilde{G}$ is more accurate on examples from $\widehat{\mathcal{M}}'$ which are closer to the original neighbor used to initialize the projection. This provides evidence that input-consistency-regularized self-training is indeed correcting the mistakes of the pseudolabeler by relying on correctly-pseudolabeled neighbors for denoising, because Figure 3 shows that examples which are closer to their neighbors are more likely to be denoised. Finally, we also remark that Figure 3 provides evidence that the denoising mechanism does indeed generalize from the self-training dataset to the population, because neither examples in $\widehat{\mathcal{M}}'$ nor their original neighbors appeared in the self-training dataset.

### E.2 PSEUDOLABELING EXPERIMENTS

In this section, we verify that the theoretical objective in (4.1) works as intended. We consider an unsupervised domain adaptation setting where we perform self-training using pseudolabels from the source classifier. We evaluate the following incremental steps towards optimizing the ideal objective (4.1), with the aim of demonstrating the improvement from adding each component of our theory:

**Source:** We train a model on the labeled source dataset and directly evaluate it on the target validation set.

**PL:** Using the classifier obtained above, we produce pseudolabels on the target training set and train a new classifier to fit these pseudolabels.

**PL+VAT:** We consider the case when the perturbation set $\mathcal{B}(x)$ in our theory is given by an $\ell_2$ ball around $x$. We train a classifier to fit pseudolabels while regularizing adversarial robustness on the target domain using the VAT loss of (Miyato et al., 2018), obtaining the following loss over classifier $F$:

$$\mathcal{L}(F) \triangleq L_{\text{cross-ent}}(F, G_{\text{pl}}) + \lambda_v L_{\text{VAT}}(F)$$

Note that this loss only enforces true stability on examples where $F(x)$ correctly predicts $G_{\text{pl}}(x)$. For pseudolabels not fit by $F$, the cross-entropy loss discourages the model from being confident, and therefore the discrete labels may still easily flip under input transformations for such examples.

**PL+VAT+AMO:** Because the theoretical guarantees in Theorem 4.3 are for the population loss, we apply the AMO algorithm of (Wei & Ma, 2019b) in the VAT loss term to regularize the robust all-layer margin (see Section 3.3). This encourages robustness on the training set to generalize better.

**PL+VAT+AMO+MinEnt:** Note that PL+VAT only encourages robustness for examples which fit the pseudolabel, but an ideal classifier should not fit pseudolabels which disagree with the ground-truth. As the bound in Theorem 4.3 improves with the robustness of $F$, we aim to also encourage robustness for examples where $F$ does not match $G_{pl}$. To this end, we modify the loss to allow the classifier to ignore $c$ fraction of the pseudolabels and optimize min-entropy loss on these examples instead. We provide additional details on how to select the pseudolabels to ignore below.

**MinEnt+VAT+AMO:** We investigate the impact of the pseudolabels by removing them from the objective. We instead rely on the following loss which simply performs entropy minimization on the target while fitting the source dataset:

$$\mathcal{L}(F) \triangleq \lambda_s L_{\text{cross-ent, src}}(F) + \lambda_t L_{\text{min-ent, tgt}}(F) + \lambda_v L_{\text{VAT, tgt}}(F)$$

We include the source loss for training stability. As before, we apply the AMO algorithm in the VAT loss term to encourage robustness of the classifier to generalize.

Table 1 shows the performance of these methods on six unsupervised domain adaptation benchmarks. We see that performance improves as we add additional components to the objective to match the theory. We note that the goal of these experiments is to validate our theory, not to push state-of-the-art for these datasets, which often relies on domain confusion (Tzeng et al., 2014; Ganin et al., 2016; Tzeng et al., 2017), which is outside the scope of our theory. For example, Shu et al. (2018) achieve strong results on these benchmarks by using a domain confusion technique while optimizing VAT loss and entropy minimization on the target while training on labeled source data. Our results for MinEnt+VAT+AMO show that when the domain confusion is removed, performance suffers and is actually worse than training on the source only for all datasets except STL-10 to CIFAR-10. We provide additional experimental details below. We use the same dataset setup and model architecture for each dataset as (Shu et al., 2018). All classifiers are optimized using SGD with cosine learning rate and weight decay of 5e-4 and target batch size of 128. The value of the learning rate is tuned on the validation set for each dataset and method in the range of values $\{0.03, 0.01, 0.003, 0.001\}$. We choose $\lambda_v$, the coefficient of the VAT loss, by tuning in the same manner in the range $\{3, 10, 30\}$. For MinEnt+VAT+AMO, we fix the best hyperparameters for PL+VAT+AMO+MinEnt and tune $\lambda_s \in \{0.25, 0.5, 1\}$ and fix $\lambda_t = 1$. We also tune the batch size for the source loss in $\{64, 128\}$. Table 1 depicts accuracies on the target validation set. We use early stopping and display the best accuracy achieved during training. All displayed accuracies are on one run of the algorithm, except for the (+MinEnt) method, where we average over 3 independent runs with the same hyperparameters.

To compute the VAT loss (Miyato et al., 2018), we take one step of gradient descent in image space to maximize the KL divergence between the perturbed image and the original. We then normalize this gradient to $\ell_2$ norm 1 and add it to the image to obtain the perturbed version. To incorporate the AMO algorithm of (Wei & Ma, 2019a), we also optimize adversarial perturbations to the three hidden layers preceding pooling layers in the DIRT-T architecture. The initial values of the perturbations are set to 0, and we jointly optimize them with the perturbation to the input using one step of gradient ascent with a learning rate of 1.

Finally, we provide details on how we choose pseudolabels to ignore for the PL+VAT+AMO+MinEnt objective. Some care is required in this step to prevent the optimization objective from falling into bad local minima. We will maintain a model whose weights are the exponential moving average of the past model weights, $F_{\text{ema}}$. Every gradient update, the weights of $F_{\text{ema}}$ are updated by $W_{\text{ema}} \leftarrow 0.999 W_{\text{ema}} + 0.001 W_{\text{curr}}$, where $W_{\text{curr}}$ is the current model weight after the gradient update. Our aim is to throw out $\tau_i$-fraction of pseudolabels which maximize $\ell_{\text{cross-ent}}(F_{\text{ema}}(x), G_{pl}(x))$, where $G_{pl}(x)$ is the pseudolabel for example $x$, and $i$ indexes the current iteration. We will increase $\tau_i$ linearly from 0 to its final value $\tau$ over the course of training. Towards this goal, we maintain an exponential moving average of the $(1 - \tau_i)$- quantile of the loss, which is updated every iteration using the $(1-\tau_i)$-quantile of the loss $\ell_{\text{cross-ent}}(F_{\text{ema}}(x), G_{pl}(x))$ computed on the current batch. We ignore pseudolabels where this loss value is above the maintained exponential moving average for the $(1 - \tau_i)$-th loss quantile.

Table 1: Validation accuracy on the target data of various self-training methods. We see that performance improves as we add components of our theoretical objective (4.1).

| Source | MNIST | MNIST | SVHN | SynDigits | SynSigns | STL-10 |
| Target | SVHN | MNIST-M | MNIST | SVHN | GTSRB | CIFAR-10 |
| --- | --- | --- | --- | --- | --- | --- |
| Source Only | 35.8% | 57.3% | 85.4% | 86.3% | 77.8% | 58.7% |
| MinEnt + VAT + AMO | 20.6% | 28.9% | 83.2% | 83.6% | 42.8% | 67.6% |
| PL Only | 38.3% | 60.7% | 92.3% | 90.6% | 85.7% | 62.0% |
| + VAT | 41.7% | 79.8% | 97.6% | 93.4% | 90.5% | 62.3% |
| + AMO | 42.5% | 81.4% | 97.9% | 93.8% | 93.0% | 63.9% |
| + MinEnt | 46.8% | 93.8% | 98.9% | 94.8% | 95.4% | 67.0% |

