# OpenReview forum: "Theoretical Analysis of Self-Training with Deep Networks on Unlabeled Data"
_ICLR.cc/2021/Conference — ICLR 2021 Oral_

### Official Review · AnonReviewer1 · 2020-10-20
**Nice paper**

**Rating:** 7
**Confidence:** 4

**Review:**

Summary of the paper: The paper gives a theoretical justification of self-training. It proposes a new notion of "expansion" - the amount of data distribution in the neighbor of an example. Here the neighbor means adding perturbations to the example, or augmentations of the example. When the label distribution satisfies nice expansion properties and that classes are properly separated according to the neighbors, the paper proves distributional guarantees of self-training. Combining with generalization bounds of DNNs, the paper also derives finite sample bounds for DNNs. The paper also verifies the expansion assumption via experiments using a GAN.

Review: Overall, the paper is written nicely and gives a novel perspective on self-training. Several questions:
	1) It is a bit strange that semi-supervised learning requires a stronger assumption on expansion than unsupervised learning. Is it possible to use the unsupervised method and then somehow align it with the semi-supervised labels?
	2) The assumptions depends heavily on $\mathcal{B}$ - if $\mathcal{B}$ is too large than we have inconsistency, if it is too small than we lose expansion. How will the results change if $\mathcal{B}$ is of other forms (e.g., coming from a model)?

I feel that the paper gives an interesting notion to consider for self-training and the statements are rigorous. I would recommend acceptance.

---

> ### Author Response · Authors · 2020-11-18
> **Response**
>
> We thank the reviewer for the positive review and insightful questions. Responses to the questions are below:
>
> --- “a bit strange that semi-supervised learning requires a stronger assumption on expansion than unsupervised learning. Is it possible to use the unsupervised method and then somehow align it with the semi-supervised labels?”
>
> This is a great observation. Indeed, the pseudolabel setting requires a larger expansion coefficient (c > 5) than unsupervised learning (c > 1), and a reasonable alternative algorithm for semi-supervised learning is to first use unsupervised learning and align the result with labeled data. However, our assumption (Assumption 4.1) for semi-supervised learning is more relaxed in another aspect --- it only requires expansion for sets with probability mass smaller than $\bar{a}$ (= the error of the pseudolabels), instead of for all sets with probability less than 0.5, which is needed in Assumption 3.2 for unsupervised learning. This difference demonstrates an advantage of using pseudolabels over unsupervised learning. In fact, in the proof in Section A (see the condition of Theorem A.4), we work with an even more relaxed assumption which only requires expansion for *all subsets of incorrectly pseudolabeled examples* and has a looser condition on c (which roughly speaking is similar to c>2). We decided to use Assumption 4.1, the version which sacrifices a bit in c, because (a) as discussed, the assumption on expansion is still weaker in another aspect, (b) we consider the exact requirement for the expansion coefficient less essential (up to constant factors), and (c) the rich literature on expansion suggests that smaller sets are more likely to have higher expansion coefficients [1]. (We discussed this after Assumption 4.1 in the paper but will clarify more in the next revision.)
>
> --- “How will the results change if B is of other forms (e.g., coming from a model)?”
>
> Our theory is quite flexible with the choice of B, though the paper focuses on the case where the transformation set B(x) is the set of points within some $\ell_2$ radius r of some augmentation of x (as described in the first paragraph of Section 3.1). As noted by the reviewer, the only requirements on B are that expansion (Assumption 3.2) and separation (Assumption 3.3) hold. The question suggested by the reviewer of defining B based on some model is a very interesting direction for future analysis, and we hope that the theory in this work may help guide us to find a better neighborhood B according to its expansion property.
>
> References:
>
> [1] Raghavendra, Prasad, and David Steurer. "Graph expansion and the unique games conjecture." Proceedings of the forty-second ACM symposium on Theory of computing. 2010.

---

> > ### Comment · AnonReviewer1 · 2020-11-20
> > **Thank you for the response**
> >
> > Thank you for the response! That resolves my questions.

---

### Official Review · AnonReviewer4 · 2020-10-28
**A strong submission that provides theory for self-training**

**Rating:** 9
**Confidence:** 4

**Review:**

**Summary:**
This paper provides a theoretical analysis of self-training for semi-supervised learning, unsupervised domain adaptation, and unsupervised learning. The authors propose a novel assumption that they dub _expansion_ to effect this analysis. The expansion assumption requires that the neighborhood of small sets have a class conditional distribution that is large. Under this assumption, the authors show population results for an algorithm that performs self-training under the objective that enforces input consistency.
They also provide finite sample guarantees based on off-the-shelf generalization bounds for unsupervised learning.

**+ves**
+ The expansion property seems neat, and seems like a natural quantity for making progress in understanding self-training theoretically
+ The authors also provide support for this empirically in Section D.
+ The paper is very well written. In particular, the summary of the theoretical results in the main paper is very well done. I also appreciated the proof intuition for Theorem 4.3's proof

**Concerns/Comments:**
- I think it would be helpful for the reader to have a brief proof sketch of the theorems immediately after the statement. The results are already summarized (well!) informally in the introduction.
- It is unclear to me how realistic Assumption 4.1 is.

**Questions for the Authors:**
- Can you expand on the optimization objective of (4.1)? How may one implement this in practice? Or, how is this analogous to what is being done in practice?
- Can you comment on the realism of Assumption 4.1? Is it possible to verify this on toy examples as you have done with the assumptions in Section 3?
- Can you comment on obtaining a finite sample version of Theorem 4.3?

---

> ### Author Response · Authors · 2020-11-18
> **Response**
>
> We thank the reviewer for the positive reviews and insightful questions. Responses to specific questions below:
>
> --- “optimization objective of (4.1)? How may one implement this in practice? Or, how is this analogous to what is being done in practice?”
>
> Both the two sums in the objective (4.1) are weighted combinations of two quantities: the input consistency regularizer, $R_B(G)$, and how well the classifier fits the pseudolabels, $L_{0-1}(G, G_{pl})$. (Note that the other quantity, $Err(G_{pl})$, doesn’t depend on the learned classifier, so it can be regarded as constant.) A practical implementation could be just linearly combining the input consistency regularizer and the pseudolabel loss with a tuning coefficient. This is closely related to recently successful self-training objectives which rely heavily on input consistency regularization [1, 2, 3]. For example, the FixMatch algorithm [2] uses strong data augmentation for the input consistency regularizer and also optimizes cross-entropy loss on the pseudolabels.
>
> --- “Can you comment on the realism of Assumption 4.1? Is it possible to verify this on toy examples as you have done with the assumptions in Section 3?”
>
> Thanks for the suggestion. Assumption 4.1 requires an expansion coefficient c > 5. For the toy distributions in Section 3, this can be achieved by increasing the $\ell_2$ radius in the definition of neighborhood; a constant factor increase would suffice. For example, in Example 3.4 for the Gaussian setting, we define the transformation set $B(x) = ${ $x’ : \|x’ - x\| \le \frac{1}{2\sqrt{d}}$}. If we increase this radius to $\frac{3}{2\sqrt{d}}$, under this new definition of neighborhood the distribution would satisfy (0.16, 6)-expansion. The same reasoning applies for Example 3.5 --- using radius of $\frac{3\kappa}{2\sqrt{d}}$ would also give (0.16, 6)-expansion.
>
> We also note that Definition A.3 provides a relaxation of Assumption 4.1 which has a looser requirement on c (which is roughly similar to c > 2) and only requires expansion for subsets of mis-pseudolabeled examples.
>
> --- “Can you comment on obtaining a finite sample version of Theorem 4.3?”
>
> Our finite sample version of Theorem 4.3 is provided in Theorem C.2 in Section C. To obtain Theorem C.2, we note that there are two population expectations which require generalization in the objective (4.1): the input consistency regularizer and the 0-1 loss on the pseudolabels. Theorems 3.7 and C.3 provide generalization bounds for these quantities which are polynomial in the weight norms and inverse margin on the training set. Plugging these generalization bounds into Theorem 4.3 gives us the finite sample version, Theorem C.2.
>
> ---”I think it would be helpful for the reader to have a brief proof sketch of the theorems immediately after the statement.”
>
> Thanks for the suggestions. We have a proof sketch in Section A.1 for Theorem 4.3 (the main theorem for the pseudolabel setting). We plan to move it back to the main body of the paper if the paper is accepted (because one more page is allowed).
>
> References
>
> [1] Berthelot, David, et al. "Mixmatch: A holistic approach to semi-supervised learning." Advances in Neural Information Processing Systems. 2019.
>
> [2] Sohn, Kihyuk, et al. "Fixmatch: Simplifying semi-supervised learning with consistency and confidence." arXiv preprint arXiv:2001.07685 (2020).
>
> [3] Xie, Qizhe, et al. "Self-training with noisy student improves imagenet classification." Proceedings of the IEEE/CVF Conference on Computer Vision and Pattern Recognition. 2020.

---

> > ### Comment · AnonReviewer4 · 2020-11-24
> > **Thanks for the response!**
> >
> > Thanks for the response. I think a brief discussion you provided for my question on Assumption 4.1 would make a nice addition to the paper if there's room!

---

> > > ### Author Response · Authors · 2020-11-24
> > > **Thanks for the suggestion!**
> > >
> > > Thank you for the great suggestion and for reading our response. We will incorporate this suggestion in the next revision of our paper.

---

### Official Review · AnonReviewer3 · 2020-10-29
**Comments**

**Rating:** 7
**Confidence:** 3

**Review:**

This work provides a unified framework to analyze the self-training, semi-supervised algorithms. The key assumptions are 1) the “expansion” assumption which characterizes the low-probability data subset must expand to a neighborhood with large probability; and 2) the neighborhoods of samples from different classes have small overlap.  Then the authors established the upper bound of the prediction error on the population when minimizing the self-training and input-consistency based loss on the population. They also extend their results to a finite-sample setting and semi-supervised setting as well.

merits
1) In the theoretical aspect, the established results can explain the success of self-training training and is of high quality.

2) Moreover, this work is well written and easy to follow.

Questions
1) Though this work provides good theoretical results, it seems that the results do not bring any practical insights to further improve the analyzed algorithms. E.g. the population-based loss in Eq. (3.4) is not used in practice, since the loss is on expectation and is relatively complex. Actually, when the network is large, then sampling data to estimate E_p[1(G(x)=y)] and R_B(G) is of the high cost.

2) About the assumption that neighborhoods of samples from different classes have a small overlap, it also may be restrictive. For example, in imagenet, there are many images that contain several objects actually. This means that the overlap between samples from different classes may be not small.

3) Finally, the key expansion constant c is not investigated on the toy and real datasets. What is the magnitude of c? this is very important since the upper bound will be large if (c/(c-1)) is hugh.

---

> ### Author Response · Authors · 2020-11-18
> **Response**
>
> We thank the reviewer for the positive review and insightful questions. Answers to specific points are below:
>
> --- “the population-based loss in Eq. (3.4) is not used in practice, since the loss is on expectation and is relatively complex.”
>
> We agree with the reviewer that the loss analyzed is not the exact one used in practice. However, the objective in (3.4) is quite similar to BYOL, one of the state-of-the-art unsupervised representation learning algorithms, in the sense that both (3.4) and BYOL encourage input consistency as the central component of the objective. Input consistency is also a key concept in SimCLR and MoCoV2, another two SOTA unsupervised learning methods. It is an interesting direction for future work to develop more practical algorithms inspired by objective (3.4). We would also like to mention that the objective (4.1) is essentially a linear combination of the pseudolabel loss and the input consistency loss (because the third term $Err(G_{pl})$ is a constant), and it captures the essence of the objective functions used in practice in semi-supervised learning [1, 2].
>
> --- “the loss is on expectation”, “when the network is large, then sampling data to estimate E_p[1(G(x)=y)] and R_B(G) is of the high cost”
>
> In Section 3.3, we provide generalization bounds between the empirical distribution and population distribution to show that the population loss can be made small by optimizing on the finite samples (see Theorems C.1 and C.2, which provide the finite sample versions of our results). The unlabeled sample cost is polynomial in the inverse margin and weights of the model. This is one of the main advantages of our work over prior approaches --- because we analyze parametric losses, it’s straightforward to obtain finite sample guarantees by using off-the-shelf generalization bounds.
>
> We’d also like to respectfully point out that the concern regarding sample cost of estimating a population objective using a finite training set arises any time we desire generalization between training and test, and is therefore not unique to self-training or the objective we analyze. Thus, this concern is to some extent orthogonal to the central focus of our work. Also, the sample complexity requirements are only for *unlabeled* data, which are typically more abundant and cheaper to obtain than labeled data.
>
> --- “assumption that neighborhoods of samples from different classes have a small overlap, it also may be restrictive. For example, in imagenet, there are many images that contain several objects actually. This means that the overlap between samples from different classes may be not small.”
>
> Thanks for the insightful comments. The authors agree that this is a valid concern, and it is an interesting direction of future work to address it. One possibility is to directly apply our theoretical results to the distribution P_single, i.e., the image distribution conditioned on only a single unambiguous class being present in the image. Prior studies of ImageNet (see Figure 4 of [3]) demonstrate that model accuracy degrades when more than one class is present in the image, so perhaps it is unrealistic to obtain strong theoretical results for such images, or it requires additional techniques to handle them better.

---

> ### Author Response · Authors · 2020-11-18
> **Response (continued)**
>
> --- “What is the magnitude of c? this is very important since the upper bound will be large if (c/(c-1)) is hugh.”
>
> We typically expect that c is on the order of a constant that is larger than 1, which means c/(c-1) is on the order of a constant. In Examples 3.4 and 3.5, we provide more quantitative guarantees by studying expansion for Gaussian distributions and Lipschitz transformations of Gaussians. For these distributions, the distance between typical points is on the order of 1, and as long as the $\ell_2$ neighborhood radius is $\Omega(1/\sqrt{d})$, it’s sufficient to achieve c = 3, so that the factor c/(c - 1)=1.5 in the unsupervised learning setting (Theorem 3.6) is small. We also note that we can increase the radius in the definition of neighborhood to achieve larger values of c, which may explain why stronger data augmentation (corresponding to larger neighborhoods) leads to better performance.
>
> Regarding finding c on real datasets: as also noted by Reviewer 2, rigorously computing c for real datasets is challenging if not impossible. The main difficulty stems from the fact that expansion is a property about the population data which the empirical distribution does not possess. It’s infeasible to check the exact expansion property of the population distribution (which we don’t have access to) for all subsets (which we cannot enumerate exhaustively).
>
> Though we can’t find c for *all* subsets, we can verify that c is large for particular subsets. In Section D, we compare the probability of the set of incorrectly pseudolabeled BigGAN images to the probability of the neighborhood of this set. We do so because our proof for semi-supervised learning only requires expansion of subsets of incorrectly pseudolabeled examples, with roughly a coefficient of c > 2. (See the condition of Theorem A.4 of Section A, which is a weaker condition than what is stated in the main body of the paper. In the main body we state the stronger assumption for simplicity of exposition.) When using an $\ell_2$ ball with radius 19.8 as the neighborhood (which corresponds to modifying each pixel by 0.024 on average out of the range [0,1]), we found an expansion coefficient c > 3, which suggests good expansion. (Concretely, the set of incorrectly pseudolabeled examples has 6% of the mass, whereas its neighbor has 20% probability mass.) More details and additional experiments are in Section D.
>
> References
>
> [1] Sohn, Kihyuk, et al. "Fixmatch: Simplifying semi-supervised learning with consistency and confidence." arXiv preprint arXiv:2001.07685 (2020).
>
> [2] Xie, Qizhe, et al. "Self-training with noisy student improves imagenet classification." Proceedings of the IEEE/CVF Conference on Computer Vision and Pattern Recognition. 2020.
>
> [3] Tsipras, Dimitris, et al. "From ImageNet to Image Classification: Contextualizing Progress on Benchmarks." arXiv preprint arXiv:2005.11295 (2020).

---

### Official Review · AnonReviewer2 · 2020-10-30
**Strong theoretical insights for a class of important but hard problems**

**Rating:** 9
**Confidence:** 4

**Review:**

In general, it is not clear, at least theoretically, how, and when unsupervised data helps to generalization of nonlinear methods. In the literature there are important and elegant works exists that analyses the impact of usage of unlabelled data during training, however, (if I am not mistaken) all these analyses have been done for linear models. Authors analyse and shed some light to several aspects of using unlabelled data during training. They formalize their analyses based on expansion assumption. I think it can be restated as the similarity between members of the same classes is bounded by below. Intuitively such an assumption is quite reasonable. The authors use the term input consistency for defining a broad set of methods e.g. transformations of the image should be similar to each other, and they also couple their analysis using the expansion assumption with input consistency. In their view input consistency brings a local stability/generalization and expansion property brings global stability/generalization. This is again quite reasonable way of thinking because intuitively just forcing an input point to be close to transformed version of itself sounds a weak property for a good generalization performance. The authors supply quite a bit of theoretical novel material to support their intuition and analysis.
Furthermore, they present some supportive experiments albeit not an extensive one.

Strong and weak points:
a)	Strong points: Please see above.
b)	The paper is quite dense, and the reader needs to be familiar with learning theory concepts. I wonder if authors would have focused on only one aspect of the problem which they are dealing. In the current version semi-supervised learning methods, unsupervised domain adaptation and unsupervised learning are covered. Every of them is a field by itself. I understand the desire of a unified and generic framework however I can imagine that there is a risk of diluting the message.
c)	Another understandably weak point is the experiments. I personally think that conducting an experimental study in the scope of this quite challenging however it will be nice see expansion property on a real dataset.
Recommendation:
Overall, I would like this paper to get published because (if I am not mistaken) paper develops an initial understanding extremely important field e.g. self-training/self-supervised learning.

Supporting arguments:
a)	I found the assumptions paper quite intuitive and necessary. Authors also supply population level guarantees for unsupervised learning. Moreover, they extend their work finite-sample guarantees by using margin concept and Lipschitz continuity. They extend their work to domain adaptation and semi-supervised learning. The novel material in the paper is extensive.


Questions:
a)	As I mentioned before I would like to expansion property on some real-world datasets. For example, can authors present some evidence of expansion property for a chosen deep neural network on a dataset (or multiple datasets) and quantify the expansion property based on some metric.
Improvement Suggestions:
a)	The paper is quite dense, and the reader needs to be familiar with learning theory concepts. I would recommend authors to decrease density of the paper and may be move some parts to the supplementary material.

Although I am quite positive about paper, I would like to see the discussion and comments. I am open to change my review to any direction if some new evidence/discussion/published work supplied.

---

> ### Author Response · Authors · 2020-11-18
> **Response**
>
> We thank the reviewer for the positive and detailed review as well as the suggestions for improvement. Our response to the reviewer’s question about experiments is below:
>
> --- “I personally think that conducting an experimental study in the scope of this quite challenging however it will be nice see expansion property on a real dataset”
>
> We thank the reviewer for the suggestions. As the reviewer noted, “experimental study [of the expansion property is] quite challenging”. The main difficulty stems from the fact that expansion is a property about the population data which the empirical distribution does not possess. It’s infeasible to check the exact expansion property of the population distribution (which we don’t have access to) for all subsets (which we cannot enumerate exhaustively). Instead, we made efforts in verifying weaker versions and direct consequences of the expansion property by assuming that the population data distribution generated by BigGAN is sufficiently close to the real distribution.
>
> First, we sanity-checked the distinction between the assumed expansion in the population distribution  and expansion in the empirical distribution (which cannot hold in high-dimension). This is demonstrated in Figure 1, where we search for neighbors w.r.t. $\ell_2$ distance of an image over the entire GAN manifold as well as the nearest neighbor on a finite sample of 100K images. The nearest neighbor from these 100K images looks nothing like the original image, whereas the neighbor in the GAN manifold is very close to the original image. This suggests the expansion is much better on population distribution than the empirical distribution (see the more detailed description in caption of Figure 1 and Section D.)
>
> Second, in Section D, we compared the probability of the set of incorrectly pseudolabeled BigGAN images to the probability of the neighborhood of this set. We do so because our proof for semi-supervised learning only requires expansion of subsets of incorrectly pseudolabeled examples. (See the condition of Theorem A.4 of Section A, which is a weaker condition than what is stated in the main body of the paper. In the main body we state the stronger assumption for simplicity of exposition.) When using an $\ell_2$ ball with radius 19.8 as the neighborhood (which corresponds to modifying each pixel by 0.024 on average out of the range [0,1]), we found an expansion coefficient c > 3, which suggests good expansion. (Concretely, the set of incorrectly pseudolabeled examples has 6% of the mass, whereas its neighbor has 20% probability mass.) Further inspection of the neighboring pairs reveals that they look quite similar, as demonstrated in Figure 1. This provides empirical support for the expansion assumption, but a full rigorous verification is computationally infeasible because it would require computing the neighborhood size of *all subsets* of incorrectly pseudolabeled examples. More details and additional experiments are in Section D.
>
> Third, another consequence of the expansion property (from classical study of expansion and isoperimetric inequalities) is that it ensures that a local random walk can traverse the manifold with a good mixing time. Qualitatively, videos publicly available on latent-space random walks for the BigGAN-generated images seem to corroborate that the manifold of each class is sufficiently connected (e.g., please see https://www.youtube.com/watch?v=YY6LrQSxIbc). (Actually, the generative model for BigGAN is quite close to the Example 3.5.)
>
> Further inspection and exploitation of variants of the expansion property is a very interesting future direction.

---

### Decision · Program_Chairs · 2021-01-07
**Final Decision**

**Decision:**

Accept (Oral)

**Comment:**

The paper looks into theoretical analysis of self-training beyond the existing linear case and considers deep networks under additional assumption on data. namely: expansion and minimal overlap in the neighborhood of examples in different classes. The results shed some light on self-training algorithms that use input consistency regularizers.
Although the assumptions are very hard to check for all input distributions, the authors make an attempt by considering output of BigGAN generator. In summary, the paper is a great first step in understanding self-training for deep networks.

The paper is overall clearly written. please add the explanation of  Assumption 4.1 as requested by Reviewer 4.

Pros: - given the extensive use of self-training the paper is of great importance to the community
-extending the analysis of self-training to deep networks
-the paper is clearly written and easy to follow

cons: -the assumptions are very hard to validate on all datasets